# Low expression of RNA sensors impacts Zika virus infection in the lower female reproductive tract

Shahzada Khan [1], Irene Lew[1], Frank Wu [1], Linda Fritts[2,3], Krystal A. Fontaine[1], Sakshi Tomar [1], Martin Trapecar [1], Hesham M. Shehata[1], Melanie Ott[1,4], Christopher J. Miller[2,3] & Shomyseh Sanjabi [1,5,6]*

Innate immune responses to Zika virus (ZIKV) are dampened in the lower female reproductive tract (LFRT) compared to other tissues, but the mechanism that underlies this vulnerability is poorly understood. Using tissues from uninfected and vaginally ZIKV-infected macaques and mice, we show that low basal expression of RNA-sensing pattern recognition receptors (PRRs), or their co-receptors, in the LFRT contributes to high viral replication in this tissue. In the LFRT, ZIKV sensing provides limited protection against viral replication, and the sensors are also minimally induced after vaginal infection. While IFNα/β receptor signaling offers minimal protection in the LFRT, it is required to prevent dissemination of ZIKV to other tissues, including the upper FRT. Our findings support a role for RNA-sensing PRRs in the dampened innate immunity against ZIKV in the LFRT compared to other tissues and underlie potential implications for systemic dissemination upon heterosexual transmission of ZIKV in women.

[1] Virology and Immunology, Gladstone Institutes, San Francisco, CA 94158, USA. [2] Center for Comparative Medicine, University of California, Davis, Davis, CA 95616, USA. [3] California National Primate Research Center, University of California, Davis, Davis, CA 95616, USA. [4] Department of Medicine, University of California San Francisco, San Francisco, CA 94143, USA. [5] Department of Microbiology and Immunology, University of California San Francisco, San Francisco, CA 94143, USA. [6] Present address: Genentech, South San Francisco, CA 94080, USA. *email: sanjabi.shomyseh@gene.com

Zika virus (ZIKV), a member of the *Flaviviridae* family, is spread by the bite of infected *Aedes* mosquitos, vertical transmission from infected mother to fetus, or sexually from one partner to another[1–4]. ZIKV infection in pregnant women has been strongly linked to incidents of birth defects, including microcephaly[5]. Clinical[6–8], epidemiological[4,9–12], and experimental[13–15] evidence all support classifying ZIKV as a sexually transmitted infection, and mathematical modeling suggests that the risk of sustained sexual transmission of ZIKV is underestimated[16]. Vaginal exposure of mice[17–19] and nonhuman primates (NHPs)[14,20] to ZIKV results in high viral replication in the vaginal mucosa. Significantly, vaginal exposure in mice during early pregnancy or sexual transmission leads to fetal brain infection and pathology[18,21,22]. We have shown that antiviral innate and adaptive immunity against lymphocytic choriomeningitis virus (LCMV-*Arenaviridae*) and ZIKV are uniquely dampened in the vaginal tissue, whereas infections of other mucosal surfaces, such as the uterus[17] or the colon[23] induce a vigorous canonical antiviral immune response. Since this diminished response in the vaginal mucosa is not common to all sexually transmitted pathogens[24,25], we hypothesized that the dampened immunity against both LCMV and ZIKV in the vaginal tissue may be due to sub-optimal innate sensing of viral RNA.

The female reproductive tract (FRT) comprises the upper FRT (UFRT) and the lower FRT (LFRT). The UFRT (uterus and ovaries) is sterile and consists of type I mucosa with a monolayer of columnar epithelial cells, whereas the LFRT (vagina and cervix) is nonsterile and consists of type II mucosa with a multilayer of squamous epithelial cells[26]. The mucosal barrier changes throughout the female menstrual cycle, which modulates the susceptibility of this tissue to infections with microbial pathogens[27]. Importantly, this difference in susceptibility is partly due to changes in estradiol-induced vaginal mucous that inhibits antigen penetration[28], as well as differences in genital mucosal permeability due to changes in epithelial thickness and expression of cell–cell adhesion molecules[29]. Accordingly, mice are resistant to vaginal infections during the estradiol-high estrus phase as the pathogens cannot penetrate the mucosal barrier, but mice become susceptible during the progesterone-high diestrus phase, most likely due to changes to the physical mucosal barrier[30–32]. Depo-provera (DMPA) is a long-acting synthetic form of progesterone, which is commonly used to make animals susceptible to vaginal infection by inducing similar changes to the mucosal barrier that naturally occur during diestrus phase, and it also prolongs the diestrus phase in mice for up to 4 weeks[30]. While achieving vaginal ZIKV infection in non-pregnant mice requires either progesterone or DMPA treatment[17,31,32], progesterone levels naturally increase during early pregnancy[33,34], and this seems to be sufficient to cause similar mucosal barrier changes to allow ZIKV to penetrate the mucosa and infect vaginal epithelium[18]. Here we opted to model these naturally high-progesterone conditions by treating the animals with low concentration of DMPA to allow ZIKV to penetrate the mucosal barrier, which is required for vaginal infection to occur in cycling mice.

Type I/III IFNs (IFNs) are central components of host defense against viral infections[35]. Viral nucleic acids stimulate pattern recognition receptors (PRRs) during infection, potently inducing IFN production. Viral RNA species are recognized by endosomal PRRs, including Toll-like receptor (TLR)3 and TLR7, and cytoplasmic receptors melanoma differentiation–associated protein 5 (MDA5) and retinoic acid–inducible gene 1 (RIG-I), which use the common downstream mitochondrial antiviral signaling (MAVS) adaptor to propagate signaling[36,37]. After detection of nucleic acids, PRRs signal through adaptors to drive expression of interferon-stimulated genes (ISGs), which provide antiviral

protection in the infected cells. Infected cells also produce IFNβ, which signals through the IFNα/β receptor (IFNAR) complex to induce the expression of ISGs in both infected and uninfected cells to initiate antiviral immunity and prevent the spread of infection. ZIKV antagonizes type I IFN signaling in human cells[38,39], allowing robust viral replication. However, ZIKV cannot inhibit type I IFN signaling in mice, and therefore cannot establish infection in wild-type (WT) mice when administered through systemic or subcutaneous routes, but can robustly replicate in *Ifnar1*$^{-/-}$ mice regardless of route of infection[40–42]. Importantly, after intravaginal (i.vag.) inoculation of DMPA-treated WT mice, we and others have found that ZIKV replicates and persists in the LFRT[17,18,43], with minimal induction of Type I/III IFNs and minimal amplification of ISGs and antiviral innate response[17]. Why ZIKV replication in the LFRT does not result in robust IFN production remains unknown.

In this study, we sought to better understand the mechanisms contributing to the dampened and delayed innate immune response against ZIKV in the LFRT. We show that limited sensing of ZIKV in the LFRT occurs around day 4 post vaginal infection. To identify the PRRs responsible for detecting ZIKV, we used knockout mice, including triple knockout (TKO) mice lacking TLR3, TLR7, and MAVS. Intriguingly, compared to WT mice, viral replication was minimally affected in the LFRT of TKO mice, although viral dissemination was more extensive and significantly more viral replication occurred in other tissues, demonstrating minimal innate protection is provided against ZIKV via RNA sensors in the LFRT. We found that low basal expression of RNA sensors in both mice and rhesus macaques is associated with dampened viral sensing in the LFRT. Furthermore, we show that while IFNAR signaling also plays minimal role in controlling viral replication in the LFRT tissue, it is essential to prevent viral dissemination to distal tissues, including the UFRT. Thus, the low expression and function of RNA-sensing PRRs in the LFRT provides a unique window of opportunity for ZIKV infection, which may lead to systemic viral spread in the absence of proper IFNAR signaling.

## Results

**T cells are required for ZIKV clearance in the LFRT**. To determine if adaptive immunity is required for ZIKV clearance after i.vag. inoculation, we infected animals with the Puerto Rican strain of ZIKV (PRVABC59, 2015) and compared viral clearance in C57BL/6 wild type (WT) and *Rag1*$^{-/-}$ mice, which lack B cells and T cells. While viral clearance occurred in WT mice by day 12 post infection (p.i.), *Rag1*$^{-/-}$ mice continued to show variable levels of viral RNA up to 30 days post infection (Fig. 1a), suggesting that adaptive immune response is required for timely viral clearance. In agreement with the role of adaptive cells in viral clearance, we detected a significant increase in the total number of B and T cells in the draining iliac lymph node (iLN) at day 6 post infection (Fig. 1b and Supplementary Fig. 1a). To determine which adaptive arm is most critical for clearing ZIKV from the LFRT, we i.vag. infected WT, CD8$^{-/-}$ (lacking CD8 T cells), MHC-II$^{-/-}$ (lacking CD4 T cells), muMT (lacking B cells), and *Rag1*$^{-/-}$ mice and compared viral RNA levels in the LFRT at day 12 p.i. While WT and muMT mice completely cleared the infection, viral RNA was detectable in most of the CD8$^{-/-}$ and MHC-II$^{-/-}$ animals (Fig. 1c). Consistent with these results, we also detected an increase in the proportion of memory (CD44+) and activated (CD44+ CD43+) CD8 and CD4 T cells at day 12 p.i. in the LFRT, iLN, and spleen of WT animals (Fig. 1d and Supplementary Fig. 1b). Collectively, these data indicate that both CD4 and CD8 T cells contribute to the timely clearance of ZIKV from the LFRT.

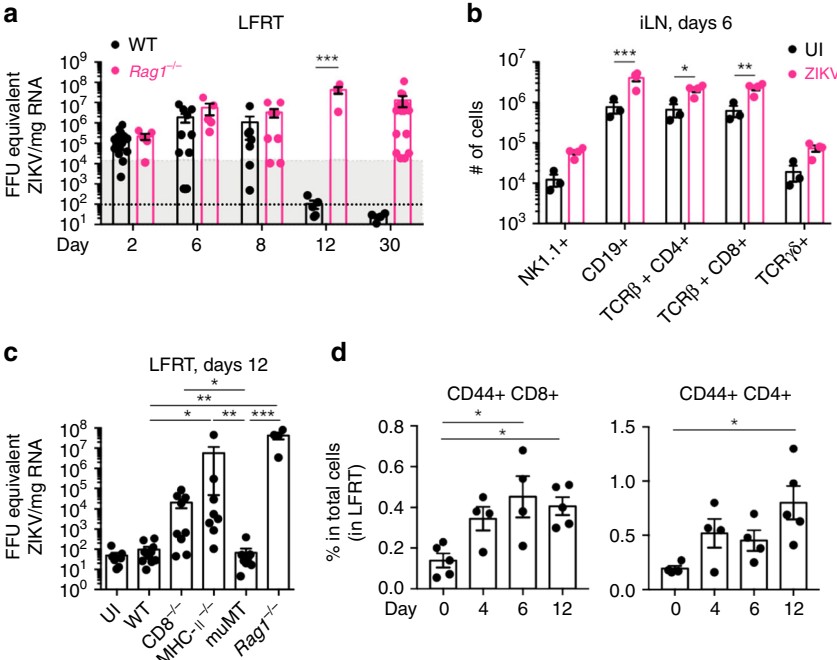

**Fig. 1** Adaptive immunity is required for resolution of vaginal ZIKV infection. Groups of DMPA-treated female mice were i.vag. inoculated with $2 \times 10^4$ FFU of ZIKV (PRVABC59). **a** FFU equivalents of ZIKV in total RNA from LFRT tissues were determined by qRT-PCR. ZIKV RNA level at day 1 post infection has been indicated by the shaded area, and the lower limit of viral detection is indicated by the lower black dotted line. WT day 2 $n = 16$, day 6 $n = 11$, day 8 $n = 7$, day 12 $n = 5$, day 30 $n = 5$; $Rag1^{-/-}$ day 2 and 6 $n = 5$, day 8 $n = 8$, day 12 $n = 4$, and day 30 $n = 17$ mice. **b** Absolute numbers of indicated immune cells in the iLN of uninfected (UI, $n = 3$ mice) and ZIKV-infected animals ($n = 4$ mice) at day 6 p.i. **c** FFU equivalents of ZIKV in LFRT at day 12 post infection. UI $n = 7$, WT $n = 11$, CD8$^{-/-}$ $n = 9$, MHC-II$^{-/-}$ $n = 8$, muMT $n = 9$, $Rag1^{-/-}$ $n = 4$ mice. **d** Frequencies of CD44+ T cells in the LFRT were determined using flow cytometry. Day 0 represents UI mice. Day 0 and 12 $n = 5$ mice each day, day 4 and 6 $n = 4$ mice each day. See Supplementary Fig. 1 for gating strategy. Data pulled from 4 **a**, or 2 **b, c** independent experiments; mean ± SEM. *$p < 0.05$; **$p < 0.01$; ***$p < 0.001$; two-way ANOVA with Bonferroni's multiple comparison test **a**, Kruskal–Wallis test with Dunn's multiple comparison **a**, **b**. Each dot represents sample from an individual mouse and each time-point represents data from a separate group of mice. LFRT, lower female reproductive tract; UI, uninfected. Source data for Fig. 1a–c are provided as a Source Data file

**Delayed sensing of ZIKV in the LFRT**. Since T cell activation is dependent on antigen presentation by antigen-presenting cells (APCs), we reasoned that APC activation must occur either in the LFRT or the draining iliac lymph node (iLN) upon viral sensing. Monocyte derived APCs play a central role in antimicrobial immune responses[44]; however, no detectable change in recruitment of various subsets of CD11b$^+$ monocytes (Supplementary Fig. 2) was observed in the first 2 days p.i. (Fig. 2a). Starting at day 3 p.i., the proportion of CD11b$^+$ cells (ly6c$^-$ IAIE$^+$) in the LFRT decreased (Fig. 2a). This was followed by a moderate induction of $Irf7$ in the LFRT (Fig. 2b), and increased expression of the APC activation markers CD86 in both LFRT and iLN and CD40 in the iLN (Fig. 2c, d). CCR2 is important for the migration of monocytes to inflamed tissues[44] and is also critical for monocyte-mediated antiviral immunity during vaginal HSV infection[45]. To determine if CCR2-mediated recruitment of monocytes is important for antiviral immunity after i.vag. ZIKV infection, we compared the kinetics of ZIKV replication and clearance in CCR2$^{+/+}$ and CCR2$^{-/-}$ mice. Consistent with the observed lack of monocyte recruitment to the WT LFRT after i.vag. ZIKV infection, we did not detect a difference in viral control or clearance in CCR2$^{+/+}$ and CCR2$^{-/-}$ animals (Fig. 2e). These data suggest that CCR2-mediated monocyte recruitment to the LFRT tissue is not required for anti-ZIKV immunity, and that ZIKV induces moderate activation of existing APCs in the LFRT. Taken together, these results indicate that in WT animals, although viral replication can already be detected by day 2 (Fig. 1a), viral sensing occurs 3–4 days p.i. (Fig. 2b) and results in

moderate APC activation 4–6 days post i.vag. ZIKV infection (Fig. 2c, d). However, this level of ZIKV sensing and innate immune activation is not sufficient to inhibit early viral replication.

**Uniquely dampened innate immunity against ZIKV in the LFRT**. To determine the contribution of RNA sensing PRRs to viral control in the LFRT and dissemination to distal tissues, we generated $Tlr3^{-/+}$ $Tlr7^{-/+}$ $Mavs^{-/+}$ mice, and then crossed them to generate WT, single, double, and triple knockout (TKO) mice, which were housed in the same animal facility. We then i.vag. infected these mice with ZIKV and compared viral loads in the LFRT, UFRT, iLN, and spleen at day 6 p.i., which is at least 2 days after $Irf7$ induction (Fig. 2b) and APC activation (Fig. 2c) in the LFRT, and prior to adaptive mediated viral clearance (Fig. 1a). In LFRT, ZIKV replication occurred efficiently regardless of the presence or absence of PRRs. ZIKV RNA copy numbers in the single knockout mice ($Tlr3^{-/-}$, $Tlr7^{-/-}$, $Mavs^{-/-}$) or $Tlr3^{-/-}$ $Tlr7^{-/-}$ double knockout mice did not significantly differ from WT mice (Fig. 3a). However, absence of $Mavs$ together with either $Tlr3$ ($Mavs^{-/-}$ $Tlr3^{-/-}$) or $Tlr7$ ($Mavs^{-/-}$ $Tlr7^{-/-}$) resulted in moderately enhanced ZIKV copy numbers in the LFRT, although the difference in viral copies between WT and $Mavs^{-/-}$ $Tlr7^{-/-}$ mice did not reach statistical significance (Fig. 3a). Compared to WT mice ([mean ± SEM] $9.0 \times 10^6 \pm 2.8 \times 10^6$ FFU equivalent/mg RNA), ZIKV replication in the LFRT was enhanced only about 1–2 log in the $Ifnar1^{-/-}$ mice ($1.6 \times 10^8 \pm 3.4 \times 10^7$ FFU equivalent/mg RNA), and to our surprise only

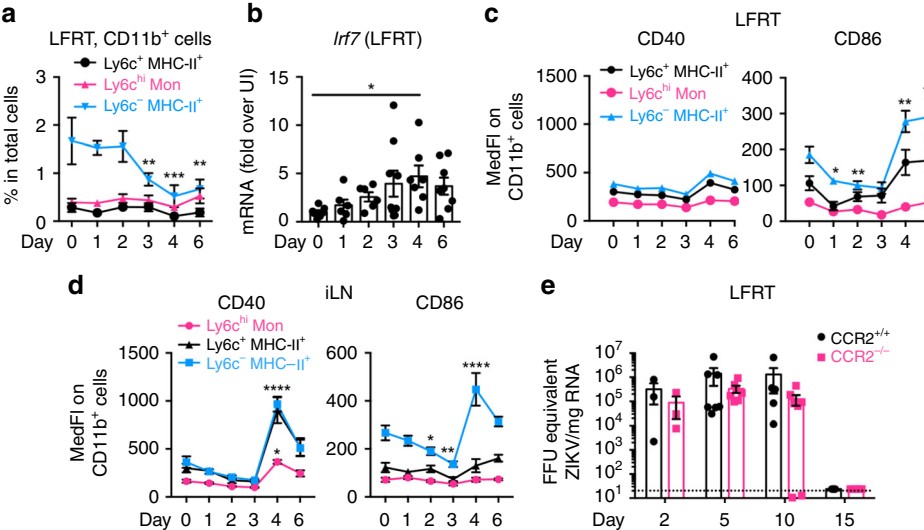

**Fig. 2** Innate immune activation is delayed after intravaginal infection with ZIKV. Groups of DMPA-treated female mice were i.vag. inoculated with $2 \times 10^4$ FFU of ZIKV (PRVABC59). **a** Frequencies of various monocyte subsets in the LFRT post-infection. CD11b$^+$ cells: CD19$^-$ TCRβ$^-$ EpCAM-1$^-$ Ly6G$^-$ CD11b$^+$ Ly6c$^{+/-}$ IAIE$^{+/-}$. Also see Supplementary Fig. 2 for gating strategy. **b** Detection of *Irf7* induction by qRT-PCR from LFRT total RNA at indicated time-points. Surface expression of activation markers on CD11b$^+$ cells in **c** LFRT and **d** iLN, as determined by flow cytometry. **e** FFU equivalents of ZIKV in total RNA from LFRT tissues of CCR2$^{-/-}$ and CCR2$^{+/+}$ animals were determined by qRT-PCR. Data pulled from two (**a**, **c**, **d**, and **e**) or three **b** independent experiments; mean ± SEM, $n = 3$ (day 2 in **e**), 3–4 (day 15 in **e**), 4 (days 4 and 6 in **a**, **c** and **d**), 6 (days 0–3 in **a**, **c**, **d**, and day 1, 2 in **b**), 7 (day 0 and 4 in **b**, day 5 and 10 in **e**), 8 (day 6 in **b**), or 9 (day 3 in **b**) mice per time-point. *$p < 0.05$, ****$p < 0.0001$; two-way ANOVA with Dunnett's multiple comparison test. Each time-point represents data from a separate group of mice, and day 0 represents data from uninfected mice. Source data for Fig. 2a–d are provided as a Source Data file

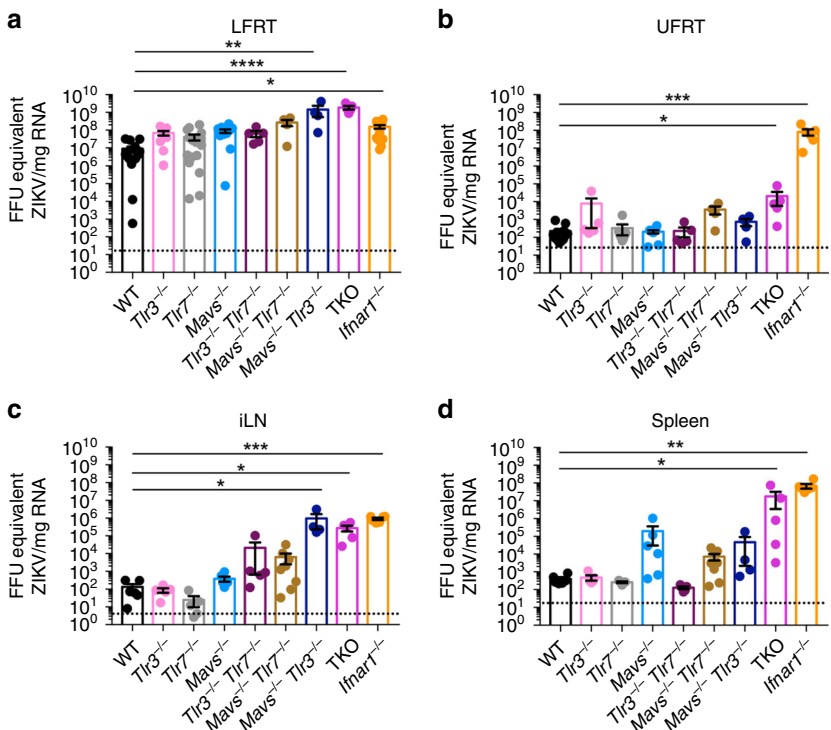

**Fig. 3** RNA sensing and IFNAR signaling control ZIKV replication and dissemination. Groups of DMPA-treated female mice were i.vag. inoculated with $2 \times 10^4$ FFU of ZIKV (PRVABC59). FFU equivalent of ZIKV detected by qRT-PCR from **a** LFRT (TKO $n = 5$, *Mavs*$^{-/-}$ *Tlr3*$^{-/-}$ $n = 7$, *Mavs*$^{-/-}$ *Tlr7*$^{-/-}$ $n = 8$, *Tlr3*$^{-/-}$ $n = 9$, and *Tlr3*$^{-/-}$ *Tlr7*$^{-/-}$ $n = 10$, *Mavs*$^{-/-}$ $n = 11$, *Ifnar1*$^{-/-}$ $n = 14$, WT and *Tlr7*$^{-/-}$ $n = 16$ mice), **b** UFRT (*Mavs*$^{-/-}$ *Tlr7*$^{-/-}$ and *Mavs*$^{-/-}$ *Tlr3*$^{-/-}$ $n = 4$, *Tlr3*$^{-/-}$, *Tlr7*$^{-/-}$ *Tlr3*$^{-/-}$ and TKO $n = 5$, *Ifnar1*$^{-/-}$ $n = 6$, *Tlr7*$^{-/-}$ $n = 8$, *Mavs*$^{-/-}$ $n = 9$, and WT $n = 12$ mice), **c** iLN (*Mavs*$^{-/-}$ *Tlr3*$^{-/-}$ $n = 4$, *Tlr3*$^{-/-}$, *Tlr7*$^{-/-}$, *Tlr7*$^{-/-}$ *Tlr3*$^{-/-}$ and TKO $n = 5$, WT, *Mavs*$^{-/-}$ and *Ifnar1*$^{-/-}$ $n = 6$, *Mavs*$^{-/-}$ *Tlr7*$^{-/-}$ $n = 8$ mice), and **d** spleen (*Mavs*$^{-/-}$ *Tlr3*$^{-/-}$ $n = 4$, *Tlr3*$^{-/-}$, *Tlr7*$^{-/-}$, *Tlr7*$^{-/-}$ *Tlr3*$^{-/-}$ and TKO $n = 5$, WT, *Mavs*$^{-/-}$ and *Ifnar1*$^{-/-}$ $n = 6$, *Mavs*$^{-/-}$ *Tlr7*$^{-/-}$ $n = 8$ mice) of i.vag. infected mice at day 6 post-infection. Data pulled from two independent experiments; mean ± SEM *$p < 0.05$, **$p < 0.01$, ***$p < 0.001$, ****$p < 0.0001$; Kruskal–Wallis test with Tukey's multiple comparison test. Each dot represents value from one mouse. Source data for Fig. 3a–d are provided as a Source Data file

about 2–3 logs in the TKO mice ($9.7 \times 10^8 \pm 4.3 \times 10^8$ FFU equivalent/mg RNA) (Fig. 3a). However, in the other tissues, ZIKV replication was strongly inhibited by the presence of sensors and IFNAR signaling (Fig. 3b–d). Significant viral dissemination to the UFRT was observed mainly in $Ifnar1^{-/-}$ and to some extent in the TKO mice (Fig. 3b). In general, IFNAR-signaling played a more significant role in inhibiting ZIKV dissemination from LFRT to other tissues compared to viral sensing (Fig. 3). Overall, these results show that both RNA sensing and IFNAR-signaling provide moderate to minimal protection in the LFRT compared to other tissues, highlighting their limited role in innate-mediated protective immunity against ZIKV in the LFRT.

**Low basal expression of RNA-sensing PRRs in the LFRT.** To assess why RNA-sensing PRRs contribute minimally to anti-ZIKV immunity in the LFRT compared to the protection provided in other tissues (Fig. 3), we compared the mRNA expression of various PRRs in the LFRT, UFRT, and iLN of uninfected mice and the corresponding tissues from rhesus macaques, which also display preferential ZIKV replication in the FRT[20]. In general, we found similar mRNA expression patterns of these sensors in both species (Fig. 4a). *Tlr3* is marginally enriched in UFRT of mice and expressed at comparable levels in all tissues in macaques, but expression of TLR3 co-receptor *Mex3B*[46] is significantly reduced in the LFRT compared to other tissues in both species (Fig. 4b), which may affect the functionality of TLR3 signaling. Compared to the iLN, the LFRT and UFRT tissues express minimal *Tlr7*. MDA5 (*Ifih1*) and RIG-I (*Ddx58*) expression are also significantly lower in the FRT than in the iLN in both species, and MDA5 expression is significantly lower in the LFRT than UFRT in mice (Fig. 4a). To determine if DMPA-treatment alters the expression of the sensors, we compared sensor expression of DMPA-treated mice with those during natural diestrus or estrus cycles. The only significant difference observed was slightly higher expression of *Tlr3* in the LFRT tissue in DMPA-treated animals (Fig. 5), which is consistent with what has previously been reported[47]. Together, these data suggest that although most RNA sensors are not hormonally regulated, their expression levels are lower in the LFRT compared to other tissues, which correlates with their limited protective function in this tissue.

**Cell-specific contribution to ZIKV sensing in the LFRT.** The LFRT is composed of hormonally regulated multi-layered squamous epithelial cells, a dense layer of stromal fibroblasts, and a dynamic population of leukocytes[27]. High progesterone levels promote polymorphonuclear neutrophil (PMN) influx into the vaginal lumen[48,49], which can provide protection against viral pathogens[50,51]. Consistently, the majority of immune cells in the vaginal lumen of DMPA-treated uninfected (Fig. 6a), and ZIKV-infected (Fig. 6b) mice are Ly6G⁺ PMNs. To determine the contribution of different cellular compartments to sensing of ZIKV in the LFRT tissue, we sorted each compartment (Supplementary Fig. 3) from WT mice at days 0, 2, 4, and 6 post i.vag. ZIKV infection. The majority of the isolated cells belong to the PMN and stromal compartments, followed by much smaller contribution from epithelium and non-PMN immune cells (Fig. 6c). We next determined the pattern of RNA sensor expression and induction in each compartment upon vaginal ZIKV infection. When normalized against the whole LFRT of uninfected mice, expression of *Ddx58* (RIG-I) and *Ifih1* (MDA5) is similar in all compartments and these genes are minimally induced upon infection (Fig. 6d). The expression of *Tlr3* and its co-receptor *Mex3B* is unaltered by infection in epithelium, stroma, and non-PMN immune cells (Fig. 6d–e). *Tlr7* expression and induction is mainly observed in immune cells (PMNs and Non-PMNs) (Fig. 6d); however, loss of TLR7 did not affect viral control (Fig. 3). Although expression of the RNA sensors may vary depending on cell-types, these data suggest that RNA sensor expression and induction is equally dampened in all cellular compartments in the LFRT tissue.

We next asked which cellular compartment in the LFRT contributed to the higher viral load in TKO or $Ifnar1^{-/-}$ mice (Fig. 3a). In WT mice, the epithelial and stroma compartments had the highest viral burden (Fig. 6f), consistent with previous reports[32,43]. Compared with WT mice, TKO mice had higher viral load in all four sorted compartments, whereas $Ifnar1^{-/-}$ mice had higher viral load only in stromal and non-PMN immune cells. These data suggest that all of these cellular compartments in the LFRT are susceptible to ZIKV infection, and while viral sensing contributes to viral control in all compartments albeit insufficiently, the protective contribution of IFNAR

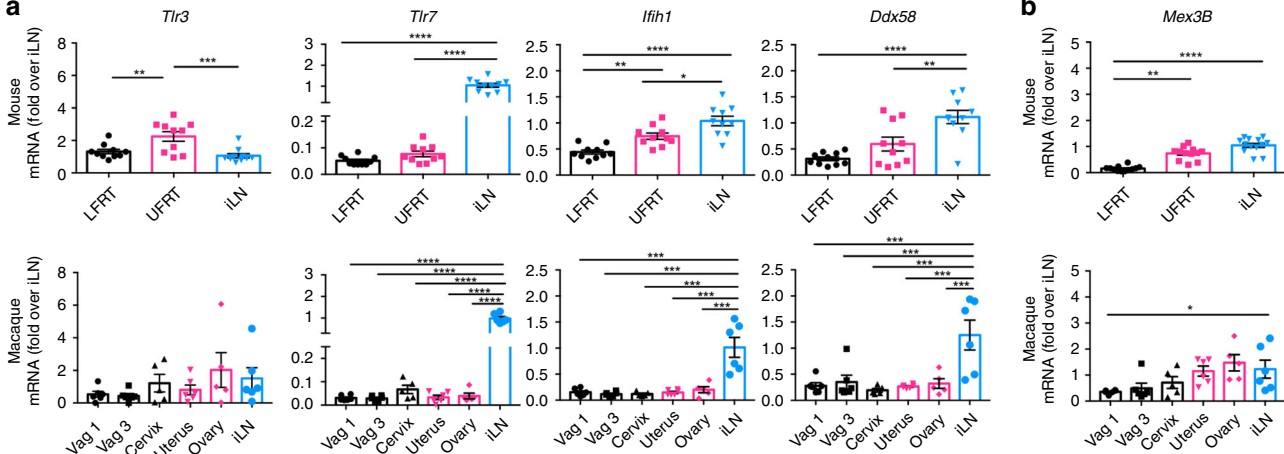

**Fig. 4** Low expression of RNA sensors in vaginal mucosa of mice and macaques. Tissues were collected from C57BL/6N female mice one week after DMPA treatment (upper panels), or from uninfected female macaques without any DMPA treatment (lower panels). Using qRT-PCR, levels of mRNA of indicated genes in all tissues were detected, normalized to GAPDH and expressed as fold-change over iLN samples. **a** Expression of RNA sensors. **b** Expression of TLR3 co-receptor. Mouse LFRT and UFRT $n = 12$, mouse iLN $n = 13$, macaque vag1, vag3, uterus and iLN $n = 6$, macaque cervix and ovary $n = 5$. Data represent mean ± SEM; *$p < 0.05$; **$p < 0.01$; ***$p < 0.001$; ****$p < 0.0001$; one-way ANOVA with Tukey's multiple comparison test. Each dot represents value from one animal. Vag, vagina (Vag 1 and Vag 3 represent two different regions of the vaginal tissue from the same animal); iLN, iliac lymph node; LFRT, lower female reproductive tract; UFRT, upper female reproductive tract. Source data for Fig. 4a–b are provided as a Source Data file

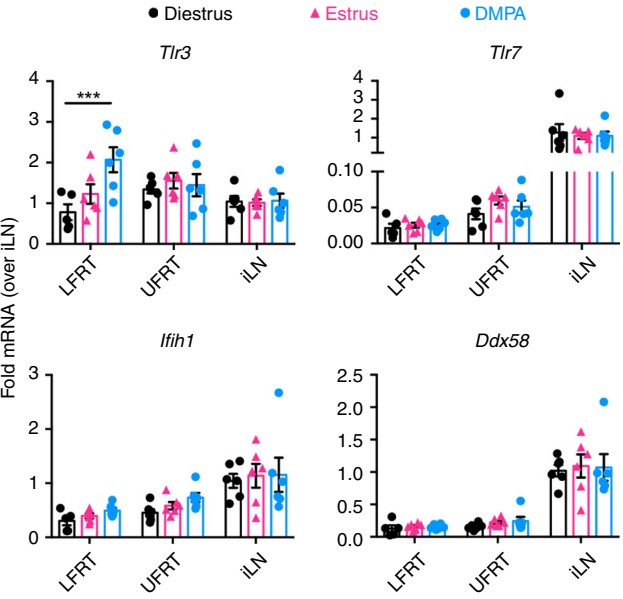

**Fig. 5** Expression of RNA sensors in DMPA-treated versus naturally cycling mice. Tissues were collected from C57BL/6N female mice one week after DMPA treatment (1.0 mg/mouse), or immediately after determining the estrus or diestrus cycle by visual monitoring of inflammation followed by microscopic confirmation of cells in the vaginal lavage. Using qRT-PCR, levels of mRNA of indicated genes in all tissues were detected, normalized to GAPDH and expressed as fold-change over iLN samples within each group. Data represent mean ± SEM; n = 6 mice per group, except n = 5 in LFRT diestrus; ***p < 0.001; two-way ANOVA with Dunnett's multiple comparison test. Each dot represents value from one animal. iLN, iliac lymph node; LFRT, lower female reproductive tract; UFRT, upper female reproductive tract. Source data for Fig. 5 are provided as a Source Data file

singling is limited to stromal and non-PMN immune cell compartments.

**IFNAR signaling limits viral dissemination from LFRT**. Our findings above indicate that viral sensing contributes only modestly to control ZIKV replication in the LFRT. To better understand this phenomenon, we measured the induction of the RNA sensors and *Irf7* in WT, TKO and *Ifnar1*⁻/⁻ mice (Fig. 7a–c). We also measured the expression of these genes in the same tissues from two different DMPA-treated macaques that were i.vag. ZIKV-infected and harvested at days 4 and 8 post infection (Fig. 7d)[20]. Dissemination of viral RNA to the UFRT and iLN was observed in all animals, albeit minimally in the WT animals (Fig. 7e). In WT mice, despite higher viral RNA in the LFRT than iLN (Fig. 7e), lower RNA senor and *Irf7* induction was detected in LFRT compared to iLN at day 4 p.i. (Fig. 7a). As expected, we observed minimal change in expression of *Ddx58*, *Ifih1*, and *Irf7* in the TKO mice, which lack RIG-I and MDA5 signaling due to deletion of MAVS (Fig. 7b). However, robust gene induction was detected in the *Ifnar1*⁻/⁻ mice (Fig. 7c) and in the macaques (Fig. 7d), but only at day 8 p.i. This similar pattern of gene induction in *Ifnar1*⁻/⁻ mice and macaques suggest that similar to human cells[38,39], it is likely that ZIKV also antagonizes type I IFN signaling in macaque cells, although further experiments with more macaques will be needed to confirm this.

To better understand the relationship between viral replication and ISG induction in each tissue, we performed linear regression analyses between viral RNA copy numbers and *Irf7* induction (Supplementary Fig. 4). In WT animals, viral RNA correlates positively with *Irf7* induction (Supplementary Fig. 4a). This

positive correlation is lost in TKO mice, suggesting that viral sensing is required for *Irf7* induction, in particular in the FRT (Supplementary Fig. 4b). Interestingly, the positive correlation between viral RNA and *Irf7* induction is lost in the iLN of *Ifnar1*⁻/⁻ mice, suggesting IFNAR signaling in addition to viral sensing is required for *Irf7* induction in the iLN. Thus, in the FRT, in the absence of IFNAR signaling, higher viral load likely results in greater induction of ISGs (Fig. 7c). This analysis also helps explain the signaling required for *Irf7* and sensor induction in the FRT versus iLN. In the iLN, ISG induction relies on both viral sensing and IFNAR signaling, whereas in the LFRT, ISG induction relies more on viral sensing than IFNAR-signaling (Supplementary Fig. 4b, c) − hence the magnitude of sensor and *Irf7* induction is less in iLN than in FRT tissue of the *Ifnar1*⁻/⁻ mice (Fig. 7c), despite high viral loads in all tissues (Fig. 7e). The similarity between gene induction in *Ifnar1*⁻/⁻ mice and macaques (Fig. 7c–d) also supports the notion that IFNAR signaling is likely antagonized in macaques as it is in humans[38,39]. Collectively, these data demonstrate that ZIKV is minimally sensed in the LFRT and can disseminate to other tissues if viral sensing or IFNAR signaling is absent or antagonized.

## Discussion

In this study, our data supports a phenomenon potentially explaining the dampened and delayed innate immunity that has been reported by several groups against vaginal ZIKV infection[17,18,20,43]. We demonstrate that in both mice and macaques, the low basal expression of known RNA-sensing PRRs and their co-receptors is associated with dampened antiviral immunity in the LFRT compared to other tissues. Our data supports a model that when the vaginal mucosa is susceptible to pathogenic infections during high progesterone conditions, the low expression of RNA sensing PRRs, which can be due to low abundance of immune cells in the LFRT, contributes to high ZIKV replication in this tissue. Furthermore, in the absence of IFNAR signaling, ZIKV dissemination can readily occur from vaginal mucosa to distal tissues. In light of the heightened rate of heterosexual transmission of ZIKV that is observed from men to women[52], the long-term persistence of ZIKV in semen[53,54], and the ability of ZIKV to antagonize IFNAR signaling in humans[38], these findings warn against the vulnerability of women during early pregnancy to heterosexual transmission of ZIKV.

The immunity in the vaginal mucosa is complicated by the physical changes that occur during the various stages of the estrous cycle[27]. This is mainly due to estradiol-induced vaginal mucous that inhibits antigen penetration[28], as well hormone-mediated changes that affect epithelial barrier thickness and permeability[29]. Interestingly however, the expression of most of the RNA sensors in the LFRT were not hormonally regulated. In a recent study, in non-pregnant ovariectomized female mice, estradiol enhanced mucosal barrier thickness and also provided protection against vaginal ZIKV infection in an IFN I/III independent manner[32], suggesting that vaginal barrier rather than viral sensing provides protection against ZIKV during the estrus phase of the cycle. In contrast, in progesterone-treated animals that support vaginal ZIKV infection, mucosal layer was thin and IFN-λ was induced at day 4 p.i., and higher viral loads were observed in mice lacking IFN-λ signaling[32]. This suggests that delayed induction of IFN-λ may provide some antiviral protection under the susceptible progesterone-high conditions. In line with these observations, in our DMPA-treated animals, we also detected moderate enhancement of APC activation and *Irf7* induction at day 4 post i.vag. ZIKV infection, which could be due to the delayed IFN-λ induction in the LFRT.

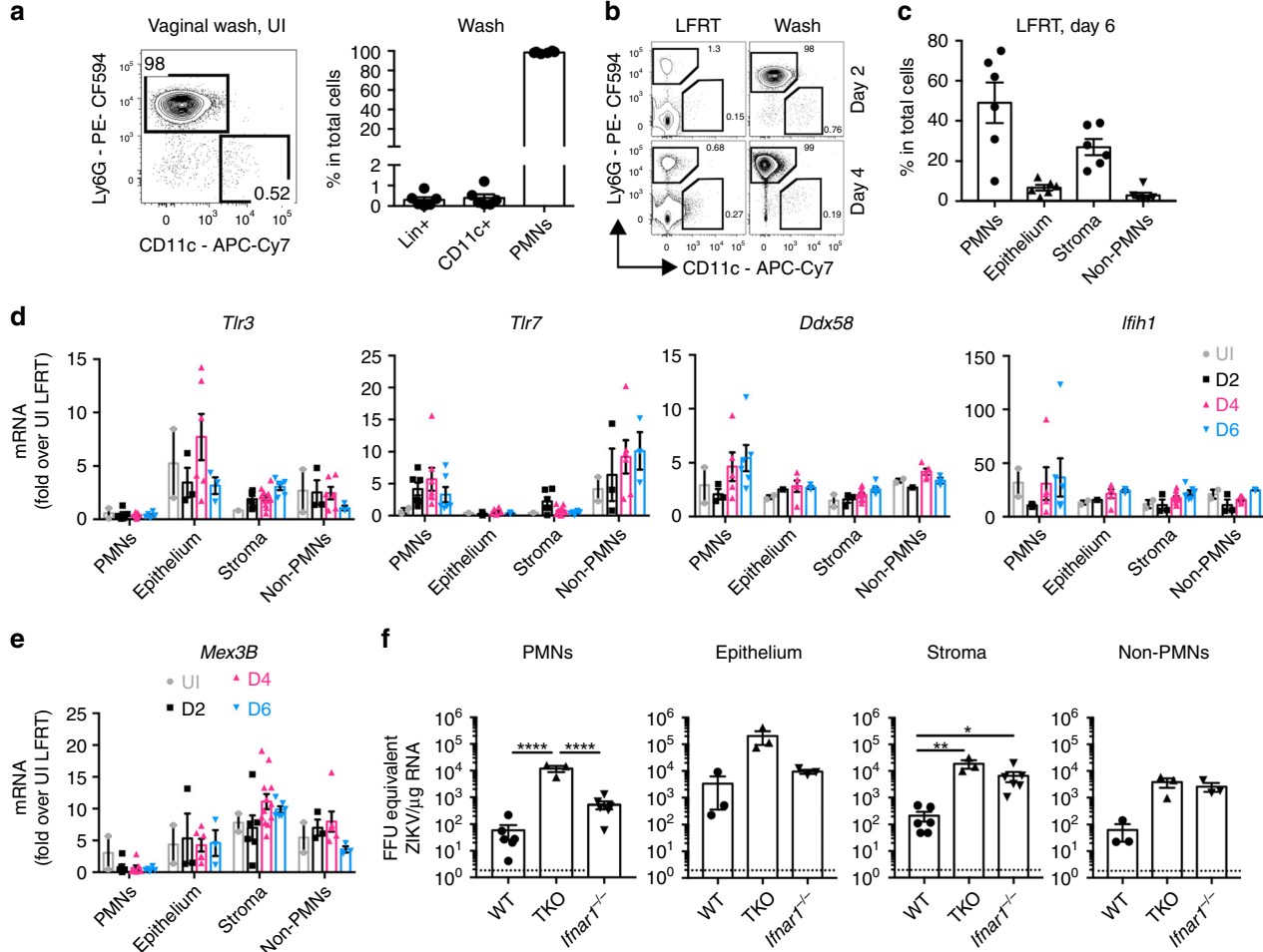

**Fig. 6** Compartmentalized role of viral sensing and IFNAR signaling in the LFRT. **a** Cells from the vaginal lumen of DMPA-treated uninfected C57BL/6N female mice were analyzed by flow cytometry. In the vaginal wash, frequencies of lineage positive (Lin+: CD19+ TCRβ+ NK1.1+), DCs (CD11c+: CD19− TCRβ− NK1.1− IAIE+ CD11c+), and polymorphonuclear neutrophils (PMNs: CD19− TCRβ− NK1.1− Ly6G+ CD11b+ SSC−) are also shown. Data represent one of four independent experiments with similar results, $n = 6$ mice. **b** Representative flow cytometry plots showing the abundance of Ly6G+ cells in the vaginal lumen (wash) compared with the LFRT tissue, at days 2 and 4 post vaginal infection with ZIKV. **c** Frequency of the four different sorted cell populations among total cells; See Supplementary Fig. 3 for sorting strategy. **d**, **e** Indicated genes were detected by qRT-PCR in total RNA from the sorted cell populations; UI $n = 4$ (sorted cells from two mice were combined for each data point), D2 and D6 $n = 6$ mice per time-point (sorted cells from two mice were combined for each data point in epithelium and non-PMNs due to low frequencies), and D4 $n = 6$ mice. **f** ZIKV RNA was detected by qRT-PCR in total RNA from the sorted cell populations; $n = 6$ mice per genotype. Due to low frequencies of epithelial cells and non-PMNs, cells from two different mice from WT or $Ifnar1^{-/-}$ mice were combined during sorting for subsequent RNA extraction and qRT-PCR. For TKO mice, cells in each population from two mice were combined together for RNA extraction. Data represent one of two independent experiments with similar results; mean ± SEM; *$p < 0.05$, **$p < 0.01$, ****$p < 0.0001$; one-way ANOVA with Tukey's multiple comparison test. UI, uninfected. Source data for Fig. 5a, c–f are provided as a Source Data file

We also found that both RNA sensing and IFNAR signaling play moderate roles in providing innate-mediated control of viral replication in the LFRT tissue. The observed ~2–3 log increase in ZIKV RNA copies in LFRT of TKO versus WT mice were due to higher viral copies in epithelial, stromal, and immune cells. This suggests RNA sensing in all of these cell types provides a moderate level of protection in LFRT, although this level of protection is not sufficient to inhibit vaginal ZIKV replication in WT mice. This is in complete contrast with the protection that viral sensing and IFNAR signaling provide in the UFRT and other distal tissues, which may be related to higher basal expression of RNA-sensing PRRs in these tissues. Compared with RNA sensing, IFNAR signaling is more critical for limiting ZIKV dissemination from LFRT to other tissues, in particular to the UFRT. In $Ifnar1^{-/-}$ mice, stromal and non-PMN immune cells in LFRT are moderately susceptible to ZIKV replication, which supports the results in an earlier study showing myeloid cells in LysMCre+ $Ifnar1^{fl/fl}$ mice may have a role in systemic dissemination of the virus after vaginal ZIKV infection[31]. Together, these data support a model where IFNAR signaling minimizes viral replication in migrating immune cells of the LFRT, limiting the ability of these cells to disseminate virus to other distal tissues. Should dissemination occur, the higher basal expression of RNA sensors in distal tissues provide the innate-mediated immunity needed to control ZIKV replication. However, under conditions where ZIKV is capable of inhibiting IFNAR signaling, such as in humans, virus may disseminate systemically.

Similar to previous findings[43,55], we found a protective role for adaptive immunity against vaginal ZIKV transmission. We showed that viral clearance is dependent on adaptive immunity, in particular T cells. However, while we show that $Rag1^{-/-}$ animals are unable to control vaginally administered PRVABC59

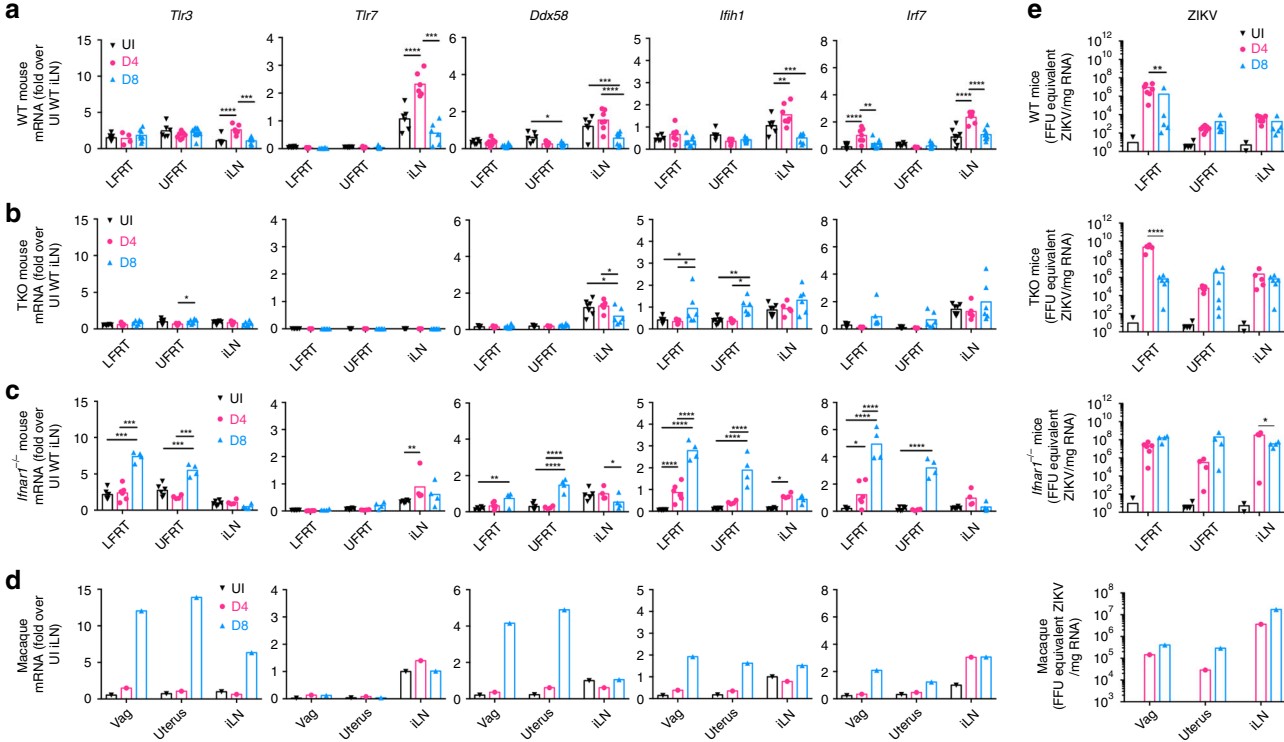

**Fig. 7** IFNAR signaling is required to limit viral dissemination from LFRT to other tissues. Female mice or macaques were i.vag. inoculated with ZIKV, and tissues were collected at day 4 or 8 post-infection. Indicated genes in total RNA were detected using qRT-PCR from (**a**) C57BL/6N WT (UI $n = 6$–7, D4 $n = 7$ except LFRT, D4 LFRT $n = 4$, D8 LFRT and iLN $n = 8$, and D8 UFRT $n = 11$ mice), (**b**) $Tlr3^{-/-}$ $Tlr7^{-/-}$ $Mavs^{-/-}$ TKO mice (UI $n = 5$, D4 $n = 5$, D8 $n = 6$ animals), (**c**) $Ifnar1^{-/-}$ mice (UI $n = 6$, D4 LFRT $n = 6$, D4 UFRT and iLN $n = 4$, and D8 $n = 4$ animals), or (**d**) macaques (UI sample was pooled from samples shown in Fig. 4 for reference, one animal each for D4 and D8), normalized to GAPDH, and expressed as fold-changes over iLN samples from naïve animals. (**e**) ZIKV RNA was detected by qRT-PCR in total RNA from the indicated tissues of WT (D4 $n = 7$ and D8 $n = 5$), TKO (D4 $n = 5$ and D8 $n = 6$), and $Ifnar1^{-/-}$ (D4 $n = 4$ and D8 $n = 4$) mice. UI WT mice ($n = 4$) were used to determine the background level of the qRT-PCR assay. All female mice were treated with 1.0 mg DMPA followed by i.vag. infection a week later. Previously described macaques were treated with 30 mg of DMPA 4 weeks prior and on the day of infection[20]. The historical ZIKV RNA levels in the macaque tissues have been re-plotted. Data represent mean ± SEM; *$p < 0.05$, **$p < 0.01$, ***$p < 0.001$, ****$p < 0.0001$; two-way ANOVA with Tukey's multiple comparison test. Each dot represent value from one animal and each time-point represents data from a separate group of animals. Vag, vagina (macaque equivalent of LFRT in mice); iLN, iliac lymph node; LFRT, lower female reproductive tract; UFRT, upper female reproductive tract; UI, uninfected. Source data for Fig. 6a–d are provided as a Source Data file

(2015) ZIKV within the first two weeks, an earlier study demonstrated that both WT and $Rag2^{-/-}$ mice could clear i.vag.-administered Cambodian FSS13025 ZIKV by about 7 days[18]. These contradictory results may be explained by potential differences in the pathogenicity of the ZIKV strains that were used in each study, or more likely, by the difference in the approach taken to monitor viral loads over time. While we monitored viral loads in whole tissue homogenate, Yockey et al. longitudinally monitored viral shedding in the vaginal lumen, which required repeated vaginal washing of the same animals over time[18]. Such repeated mucosal disturbance may cause the activation of an altered innate immune response in the LFRT tissue[51], resulting in enhanced viral control. These findings also point to the complexity of how innate and adaptive immunity is elicited in the FRT against sexually transmitted pathogens, where in addition to the type of pathogen, the effect of the mucosal disturbance should also be considered during study design and data interpretation.

A few studies have shed light on which RNA-recognizing PRRs can detect ZIKV in different tissues. In human foreskin fibroblasts, ZIKV RNA accumulation is enhanced upon silencing of $Tlr3$, $Ddx58$ and $Ifih1$, but not $Tlr7$[56], suggesting a protective role for TLR3, RIG-I, and MDA5 in skin fibroblasts. However, it was recently also demonstrated that ZIKV RNA genome is recognized by RIG-I but not MDA5, and only the RNAs that co-purified with RIG-I during ZIKV infection were shown to be immune-stimulatory[57]. Interestingly, RIG-I signaling can potently restrict ZIKV replication in primary human DCs, while IFNβ signaling is less effective[39]. In contrast, TLR3 activation via ZIKV infection in neuronal organoids has been shown to drastically change the expression profile of genes related to cerebral development and induce apoptosis and neurogenesis, while treatment of ZIKV-infected organoids with a specific TLR3 inhibitor partially reverses the growth attenuation[58]. Collectively, these studies suggest that ZIKV RNA can be sensed by TLR3 and RIG-I, and the downstream effect is cell-type specific.

A gross comparison of the mRNA expression of the RNA sensors in the LFRT, UFRT, and iLN demonstrated that the expression of these sensors and/or their co-receptor was significantly lower in the LFRT compared to the iLN, a phenomenon that we found not to be hormonally regulated, yet conserved from mice to macaques, and perhaps even more pronounced in macaques. It will be important to determine if this pattern of sensor expression is also conserved in humans, and how exactly it is regulated. This pattern of lower level of sensor expression in the LFRT compared to other tissues may be important for reproductive biology. For example, semen and semen exosomes contain high amounts of RNA that can provide inheritance of acquired epigenetic traits[59,60]. It may thus be evolutionarily more advantageous not to elicit unnecessary innate immunity against male semen in the vaginal mucosa; even though this creates a

loophole in immunity and a window of opportunity for RNA viral pathogens to be sexually transmitted.

Although mice and macaques showed similar patterns of sensor expression in uninfected animals, the two species displayed differential patterns of induction after vaginal ZIKV infection. In mice, minimal induction was mainly observed in the iLN at day 4 p.i., which was consistent with the timing of APC activation. However, in the two DMPA-treated infected macaques we analyzed, we detected a robust induction in the FRT at day 8 p.i. Although further macaque studies will be needed to confirm these results, from these preliminary findings we hypothesize that the delay in sensor induction may be related to the potential ability of ZIKV to inhibit IFNAR signaling in macaques, but not in WT mice. In support of this hypothesis, we saw that $Ifnar1^{-/-}$ mice had a very similar delayed pattern of sensor and $Irf7$ induction in the FRT after ZIKV infection. Importantly, the similar delayed patterns of gene induction that is seen in macaques and $Ifnar1^{-/-}$ mice suggests that innate immunity against ZIKV in human vaginal tissues after heterosexual transmission may also be highly delayed, as ZIKV can inhibit IFNAR signaling in human cells[38,39]. A delayed protective immune response in the vaginal tissue may increase the risk of fetal disease if infection occurs during pregnancy. Indeed, it was recently reported that fetal infection and pathology in AG129 mice that lack type I/II IFN signaling is more common and severe after sexual ZIKV transmission compared to subcutaneous ZIKV transmission[22]. Furthermore, at day 8 p.i. we also detected sporadic induction of $Ifih1$ and $Irf7$ in the TKO animals, which lack the classical RNA sensing PRRs. This could be due to high levels of ZIKV replication, which can lead to NS5-mediated inflammasome activation and IL-1beta-mediated induction of inflammation, as was recently described[61].

In conclusion, our findings show that the expression levels of RNA sensors are significantly lower in LFRT compared to other tissues at both progesterone–high and –low cycles. This low expression of RNA-sensing PRRs in the LFRT correlates with minimal protection against infection with ZIKV, particularly when mucosal barrier functions are relatively compromised during progesterone-high conditions. These observations regarding the unique features of innate immunity in the LFRT should help inform future efforts toward eliciting a protective immune response against ZIKV infection, as well as potentially other RNA viral pathogens.

## Methods

**Mice and macaques**. C57BL/6NCr WT (NCI), Rag1[tm1Mom] ($Rag^{-/-}$)[62], Tg (TcrLCMV)327Sdz[63], H2[dlAb1–Ea] (MHC-II[−/−])[64], Igh-6[tm1Cgn] (muMT[−/−])[65], and Cd8a[tm1Mak] (CD8[−/−])[66] were used in these studies. B6 backcrossed $Tlr3^{-/-}$ (JAX), $Tlr7^{-/-}$ (JAX), $Ifnar1^{-/-}$ (JAX), and $Mavs^{-/-}$ mice from Dr. James Chen (UT Southwestern) were kindly provided by Dr. Akiko Iwasaki (Yale University). We crossed $Mavs^{-/+}$ $Tlr3^{-/+}$ $Tlr7^{-/+}$ females to $Mavs^{-/+}$ $Tlr3^{-/+}$ $Tlr7^{-/+}$ males to generate the experimental $Mavs^{+/+}$ $Tlr3^{+/+}$ $Tlr7^{+/+}$ (WT), $Mavs^{-/-}$ $Tlr3^{+/+}$ $Tlr7^{+/+}$ ($Mavs^{-/-}$), $Mavs^{+/+}$ $Tlr3^{-/-}$ $Tlr7^{+/+}$ ($Tlr3^{-/-}$), $Mavs^{+/+}$ $Tlr3^{+/+}$ $Tlr7^{-/-}$ ($Tlr7^{-/-}$); $Mavs^{-/-}$ $Tlr3^{-/-}$ $Tlr7^{+/+}$ ($Mavs^{-/-}$ $Tlr3^{-/-}$), $Mavs^{-/-}$ $Tlr3^{+/+}$ $Tlr7^{-/-}$ ($Mavs^{-/-}$ $Tlr7^{-/-}$), $Mavs^{+/+}$ $Tlr3^{-/-}$ $Tlr7^{-/-}$ ($Tlr3^{-/-}$ $Tlr7^{-/-}$), and $Mavs^{-/-}$ $Tlr3^{-/-}$ $Tlr7^{-/-}$ (triple knockout, TKO) animals. All animal experiments were conducted with all relevant ethical regulations for animal testing and research and were done in accordance with guidelines set by the Institutional Animal Care and Use Committee of the University of California, San Francisco under protocol #AN151865–03A.

The rhesus macaques used in this study were part of a previously published study on vaginal ZIKV transmission[20]. Briefly, captive-bred mature (> 5year old) parous, cycling female rhesus macaques (Macaca mulatta) used in this study were from the California National Primate Research Center. All animals were negative for antibodies to WNV, HIV-2, SIV, type-D retrovirus, and simian T cell lymphotropic virus type 1 at the time the study was initiated. The animals were housed in accordance with the recommendations of the Association for Assessment and Accreditation of Laboratory Animal Care International Standards and with the recommendations in the Guide for the Care and Use of Laboratory Animals of the National Institutes of Health. The Institutional Animal Use and Care Committee of

the University of California, Davis, approved these experiments under protocol # 19471. When immobilization was necessary, the animals were injected intramuscularly with 10 mg/kg of ketamine HCl (Parke-Davis, Morris Plains N.J.). All efforts were made to minimize suffering. Details of animal welfare and steps taken to ameliorate suffering were in accordance with the recommendations of the Weatherall report, "The use of non-human primates in research". Animals were housed in an air-conditioned facility with an ambient temperature of 21–25 °C, a relative humidity of 40–60% and a 12 h light/dark cycle. Animals were individually housed in suspended stainless steel wire-bottomed cages and provided with a commercial primate diet. Fresh fruit was provided once daily and water was freely available at all times. A variety of environmental enrichment strategies were employed including housing of animals in pairs, providing toys to manipulate and playing entertainment videos in the animal rooms. In addition, the animals were observed twice daily and any signs of disease or discomfort were reported to the veterinary staff for evaluation. The animals were sacrificed by intravenous administration of barbiturates.

**Viral infection**. ZIKV strain PRVABC59 (Puerto Rico, 2015) was purchased from the American Tissue Culture Collection (ATCC, VR-1843). Stocks of ZIKV were propagated in Vero cells (ATCC, CCL-81) and viral titers were determined by focus-forming assay on LLC-MK2 cells (ATCC, CCL-7). Female mice were injected subcutaneously with Depo-Provera (Amersham Pharmacia/Upjohn) at 1 mg per mouse in a 200-μl volume in PBS, 7 days prior to infection. On day 7, mice were i.vag. infected with $2.0 \times 10^4$ FFU of ZIKV. For i.vag. infection, mice were anesthetized using isofluorane and 20 μl of viral suspension was inoculated into the vaginal cavity using a p200 micro-pipette, without causing abrasion or removal of the mucous layer.

The atraumatic vaginal virus inoculation procedure in macaques consisted of inserting a 1 CC needless tuberculin syringe containing 1 ml of the ZIKV stock ($10^7$ PFU/mL) into the vagina until the tip touched the cervix. Then the syringe was gently withdrawn while the viral inoculum was expelled. Animals that remained uninfected after 8 vaginal ZIKV inoculations were treated with DMPA. Briefly, 4 weeks before, and on the day of, challenge with ZIKV, 30 mg of DMPA was administered by intramuscular injection[20].

**Determining the stage of natural estrous cycle**. To determine the stage of natural estrous cycle, animals were examined visually for vaginal inflammation and vaginal smears were collected by gentle lavage with 50 μl of sterile PBS using a sterile 200-μl pipet tip. Stage of estrous cycle was confirmed by examining the proportion and morphology of leukocytes and epithelial cells present under a light microscope[67].

**Isolation of mouse immune cells**. Lymph node and spleen tissues were processed into single-cell suspensions. Vagina was separated from urethra and cervix. The LFRT consisted of vaginal tissue and the transformation zone. LFRT or UFRT tissues from individual mice were digested in 1 mg/ml collagenase type IV (Worthington) and 75 μg/ml DNase I (Roche) in RPMI medium. Single-cell suspensions were obtained from the digested tissues using GentleMACS Dissociator (Miltenyi Biotec) according to the manufacturer's protocol. Cell counts were obtained using an Accuri cytometer (BD), and cell numbers were normalized before antibody staining.

**Flow Cytometry**. Single-cell suspensions were first stained with Aquamine Live/dead dye (Invitrogen) according to manufacturer's instructions. After blocking Fc receptors with anti-CD16/CD32, cells were incubated with 40 μl of a mixture of fluorescence-conjugated anti-mouse antibodies for 40 min at 4 °C. Stained cells were washed once and acquired with an LSR II flow cytometer and FACSDiva software (BD). For some experiments, LFRT cells were sorted using BD FACSAria II flow cytometer. Percp-Cy5.5 CD4 (clone GK1.5, cat# 100434), PE CD43 (clone 1B11, cat# 121208), AF700 IAIE (MHC-II; clone M5/114.15.2, cat# 107622), PE-Cy7 Ly6c (clone HK1.4, Cat# 128018), A647 EpCAM-1 (epithelial cell adhesion molecule 1; clone G8.8, cat# 118212), and APC-Cy7 CD11c (clone N418, cat# 117324) antibodies were from BioLegend. PE-CF594 Ly6G (clone 1A8, cat# 562700), PerCP-Cy5.5 CD11b (clone M1/70, cat# 550993), BV605 CD44 (clone IM7, cat# 563058), BV605 CD86 (clone GL-1, cat# 563055), PE-CF594 TCRβ (clone H57–597, cat# 562841), PE CD45 (clone 104, cat# 560695), FITC CD45 (clone 104, cat# 553772), and biotin-conjugated NK1.1 (clone PK136, cat# 553163) and TCRβ (clone H57–597, cat# 553169) antibodies were from BD Biosciences. Biotin-conjugated CD19 (clone 1D3, item# AM005) and AF700 CD8α (clone YTS169.4, item# AM024) antibodies were from the University of California, San Francisco Monoclonal Antibody Core. All antibodies were used at a final dilution of 1:1500, except APC-Cy7 CD11c, Percp-Cy5.5 CD4 and PerCP-Cy5.5 CD11b were used at a dilution of 1:1000.

**Quantitative real-time PCR**. Total RNA from tissues were isolated using TRIzol reagent and cDNA was synthesized using Maxima First Strand cDNA Synthesis kit with dsDNAse (Thermo Fisher Scientific). ZIKV RNA levels and gene expressions were determined by quantitative reverse transcriptase PCR (qRT-PCR) using 2x SensiFAST probe Hi-Rox mix (Bioline) with gene-specific primers[17]. Viral burden

was determined from a standard curve produced using serial 10-fold dilutions of ZIKV RNA. Primer sequences are listed in Supplementary Table 1. For determining relative mRNA expression of PRRs across different tissues, the average $\triangle Ct_{iLN}$ value ($Ct_{gene} - Ct_{GAPDH}$) from all uninfected iLNs was used to calculate $\triangle\triangle Ct$ value ($\triangle Ct_{tissue} - \triangle Ct_{iLN}$) of genes in individual tissues of each mouse, which gave fold induction of genes in tissues over the iLN values. Similarly, for sorted cells, mRNA expression levels of each gene were normalized to the whole LFRT from uninfected mice.

**Statistical analysis**. GraphPad Prism 6.0 software was used for data analysis. Statistical significance was determined by Kruskal–Wallis analysis followed by Dunn's multiple comparison test or one-way ANOVA with Dunnett's multiple comparison test for data with a single variable, and by two-way ANOVA with Tukey's or Dunnett's multiple comparison test for data with two variables. All experimental data were included in the statistical analyses.

**Reporting summary**. Further information on research design is available in the Nature Research Reporting Summary linked to this article.

## Data availability
The authors declare that the data supporting the findings of this study are available within the paper and its supplementary information file. The source data underlying Figs 1a–d, 2a–e, 3a–d, 4a–b, 5, 6a, 6c–f, 7a–e and Supplementary Fig. 4a–c are provided as a Data Source file.

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

## Acknowledgements

This work was supported by support from the Gladstone Institutes and an NIH DP2 AI112244 to S.S., NIH R01 AI097552 to M.O., and NIH R21 OD023818 to C.J.M. This publication was made possible with help from the UCSF-Gladstone CFAR, an NIH-funded program (P30 AI027763) supporting Gladstone's flow core and NIH S10 RR028962 and the James B. Pendleton Charitable Trust for the FACSAria cell sorter. The Gladstone Institutes received support for its animal care facility from a National Center for Research Resources Grant RR18928.

## Author contributions

Conceptualization, S.S. and S.K.; Methodology, S.K., I.L., F.W., L.F., M.T., and H.M.S.; Formal Analysis, S.S., S.K., I.L., and F.W.; Resources, M.O., K.A.F., S.T., and C.J.M.; Writing – Original Draft, S.S. and S.K.; Writing – Review & Editing, S.S., S.K., M.T., H.M.S., M.O., C.J.M., F.W., K.A.F., S.T.; Visualization, S.S., S.K., I.L., and F.W.; Supervision, S.S. and S.K.; Funding Acquisition, S.S., M.O., and C.J.M.

## Competing interests

The authors declare no competing interests.
