## [Peer Review File · Nature Communications]

Reviewers' comments:

Reviewer #1 (Remarks to the Author):

This manuscript by Khan et al. provides experimental evidence that low steady-state expression of RNA sensors in the lower female reproductive tract (LFRT) are likely responsible for reduced and delayed innate immune responses after vaginal Zika virus (ZIKV) infection. Using a panel of knockout mice (single, double, and triple knockouts) and tissues from ZIKV i.vag inoculated macaques, they demonstrate that low basal expression of RNA sensors in both mice and macaques is associated with dampened innate-mediated control of viral replication in the lower female reproductive tract. They also demonstrate that IFNAR signaling is not critical for virus control in the LFRT but it is important for preventing viral dissemination to other tissues. Based on these results, the authors conclude that in humans, sexual transmission of ZIKV could lead to systemic dissemination, including to the fetoplacental unit. The results are interesting and there is substantial enthusiasm for the conclusions drawn, especially with the additions to the manuscript since the prior submission.

Minor comments:

Figure 1 and throughout. I suggest not using green and red because it is difficult for those with color blindness to differentiate the data points.

Figure 1A, please include an explanation for the dashed lines in the legend.

Figure 3. The error bars are difficult to see. Perhaps you can make them black instead of the same color as the data points.

Line 150-154: I am confused about the standard deviations that are listed here. They don't appear to be correct. Please double check.

Throughout: Ifnar^{-/-} mice should be Ifnar1^{-/-} mice and there is no description of the provenance of these animals in the methods.

Reviewer #2 (Remarks to the Author):

All of my comments have been adequately addressed. I, Shannan Rossi, reviewed this manuscript. I feel the changes you made have significantly strengthened this manuscript.

Reviewer #3 (Remarks to the Author):

The study by S Khan et al provide insights into different aspects during infection in the lower female reproductive tract. Interestingly, viral replication in the FRT and dissemination to other tissue occurs during the progesterone high phase. How this picture would look in a progesterone-low phase has unfortunately not been studied. Therefore, conclusions how this phase influences RNA levels, how these levels influence ZIKV infection cannot be made. The general observation that RNA sensors are expressed at different levels in different tissues and that expression in lymphatic tissue is higher than in epithelial tissue does not explain how ZIKV is or is not sensed and what enables it to spread. All in all, the data is not enough to substantiate the conclusions made. Vice versa the data are so extensive that clear correlations cannot be seen. Many of the results could be interpreted differently, mostly because the final experimental prove of “sensing” or “dissemination” or “RNA expression and IFN signaling” is missing and much is left to hypothesizing.

Specific comments:

Abstract: To make the conclusions basal expression basal levels and sensing in DMPA treated vs untreated mice should be checked. And correlated to replication and dissemination.

Line 55 Are macaques more susceptible to ZIKV infection during the progesterone-high phase?

Line 79 How is this in macaques?

Fig 2 b is the increase statistically significant?

Line 123 Which cells are responsible for the Irf7 increase if the CD11b subset decreases? If those cells leave, have they sensed the virus and migrate to the lymphnodes? Like in Fig 5 the cellular compartments could be checked for Irf7. And what about the increase in other sensing markers like cytokines/IFN?

Line 126 Not only CCR2 mediates Monocyte migration.

Line 133 The main conclusion would be: monocyte migration to the tissue (and ccr2 signaling) is not needed for sensing. Could be other (resident) cells.

Line 134 How do these results show that the sensing is dampened and delayed? Or that the APC activation is moderate? Compared to what, other tissues, other cells, other viruses, progesterone-low mice? What is normal?

Line 136 Where is it shown that T cells are primed 4-6 days post infection? Fig 1 only shows 4 dpi, and Fig S2 only shows 6 and 12 dpi, and Fig 2 doesn't show T cell priming?

Line 146 The RNA copy numbers between KO mice and WT mice do differ about a log.

Line 154 "ZIKV replication was strongly inhibited by the presence of sensors" ... indicating that the sensing is not dampened?

Fig 2. To conclude that sensing is delayed or dampened, more direct sensing markers need to be evaluated, and a fair control group which shows "normal" sensing included.

Line 160 "RNA sensing and IFNAR signaling provide moderate to minimal protection in the LFRT compared to other tissues" How can this be concluded if the no experiment has been done where other tissues were inoculated with ZIKV and sensing and dissemination monitored there? Isn't simply the site of inoculation the site where replication is most prominent (and sensing and therefore IFN signaling occurring first to minimize spreading)?

Line 164 Again, maybe simply the site of injection decides for the high replication? The protection in other tissues might only be due to delayed viral kinetics (simple time and anatomy) but a preactivated immune system? Vice versa, if the skin/liver/blood would be infected first, would the virus spread faster to the LFRT? Or not at all because there the sensors are more effective?

Line 169 TLR3 is marginally enriched

Line 170 why are the macaques not DMPA treated? Would be more comparable with Fig 6d

Line 176 The conclusion that epithelium has a dampened immunity due to the fact that it has less sensors expressed than in an (always slightly activated) lymphoid tissue does not seem fair. Which non-lymphatic tissue shows the same activation as lymph node and therefore has "not-dampened" immunity?

Line 186 Only the induction of sensors by different cell compartments is measured. This does not explain which compartment is responsible for the first sensing, simply which compartments responds to a already sensed infection by increasing sensing themselves.

Line 198 If all compartments show dampened RNA sensor expression, how is this in other tissues?

Line 200 If IFNAR signaling in the LFRT is examined, why not measure IFN levels? Especially IFN III levels and signalling should be focused on as this type is important in mucosal tissues. IFNAR -/- mice of course are a nice tool, but it only focuses on type I and II IFN and might not explain sensing and replication in the FRT.

Fig 5f. This Fig like Fig 3a shows that sensing in the FRT in wt mice is strong and reduces replication of the virus.

Line 206 how is the sensing insufficient if the viral loads are decreased?

Fig 6 Comparing the sensors within the different tissues would be interesting. Which sensors are highest expressed in the different tissues? And then compare uninfected vs infected. And compare the induction of sensors compared to the specific tissue. It looks like the induction in the vaginal tissue is higher than in the ILNs

Fig 6d The induction (compared to basal UI levels) of sensors in the LFRT is always higher than in the iLNs

Line 226-240 this parts needs to be easier to grasp.

Fig 6E This Figure shows that sensing at the FRT leads to lower dissemination. If sensing or IFN signaling is decreased more virus spreads throughout the system. Doesn't this argue for the fact that the sensing in the FRT is efficient?

Line 242 The data does not back up this general conclusion. Collectively, these data demonstrate that ZIKV is already sensed in the FRT, but has the ability to disseminate (in macacaques mybe due to IFN antagonization and in mice only efficiently if IFN or sensing is inactivated).

Line 245 "mechanism" is overstated

Line 248 the association seem weak. More other tissues that are not lymphatic are needed.

Line 250 why the low RNA expression and not the low availability of APCs/ T cells/ B cells? From the data it is not conclusive that the RNA expression is responsible for replication.

Line 265 could IFN-L induction be measured?

Line 275 or the fact that ZIKV has been sensed in the LFRT and needs to disseminate from there, which is dampened due to its activation and IFNAR signaling?

Line 282 or the disemminating APCs already prime the distal tissue so that the virus cannot disemminate properly

Reviewer #4 (Remarks to the Author):

I appreciate the author's attention to the comments of all four referees that provided comments on the manuscript. Significant changes were made to the manuscript that in many instances improve clarity and context. My appreciation of the work was enhanced by the author's rebuttal and a second look at the paper. No critical issues remain.

Reviewer #1 (Remarks to the Author):

This manuscript by Khan et al. provides experimental evidence that low steady-state expression of RNA sensors in the lower female reproductive tract (LFRT) are likely responsible for reduced and delayed innate immune responses after vaginal Zika virus (ZIKV) infection. Using a panel of knockout mice (single, double, and triple knockouts) and tissues from ZIKV i.vag inoculated macaques, they demonstrate that low basal expression of RNA sensors in both mice and macaques is associated with dampened innate-mediated control of viral replication in the lower female reproductive tract. They also demonstrate that IFNAR signaling is not critical for virus control in the LFRT but it is important for preventing viral dissemination to other tissues. Based on these results, the authors conclude that in humans, sexual transmission of ZIKV could lead to systemic dissemination, including to the fetoplacental unit. The results are interesting and there is substantial enthusiasm for the conclusions drawn, especially with the additions to the manuscript since the prior submission.

Authors: We appreciate the reviewer's enthusiasm for our manuscript and are thankful for their critical comments that helped us improve/clarify the description of our results and thus the conclusions in our revised manuscript.

Minor comments:

Figure 1 and throughout. I suggest not using green and red because it is difficult for those with color blindness to differentiate the data points.

Authors: Green and red colors have been replaced with alternatives.

Figure 1A, please include an explanation for the dashed lines in the legend.

Authors: We have now added the explanation in the figure legend.

Figure 3. The error bars are difficult to see. Perhaps you can make them black instead of the same color as the data points.

Authors: We have changed the color of all the error bars to black.

Line 150-154: I am confused about the standard deviations that are listed here. They don't appear to be correct. Please double check.

Authors: This data is presented in log scale, and due to the spread in viral loads, the SD values are quite large, we therefore substituted these values with SEM values.

Throughout: Ifnar^{-/-} mice should be Ifnar1^{-/-} mice and there is no description of

the provenance of these animals in the methods.

Authors: We have replaced *Ifnar*^{-/-} with *Ifnar1*^{-/-} throughout, and have added description of provenance of these mice in the methods section.

Reviewer #2 (Remarks to the Author):

All of my comments have been adequately addressed. I, Shannan Rossi, reviewed this manuscript. I feel the changes you made have significantly strengthened this manuscript.

Authors: We appreciate Dr. Rossi's support for publication of our revised manuscript.

Reviewer #3 (Remarks to the Author):

The study by S Khan et al provide insights into different aspects during infection in the lower female reproductive tract. Interestingly, viral replication in the FRT and dissemination to other tissue occurs during the progesterone high phase. How this picture would look in a progesterone-low phase has unfortunately not been studied. Therefore, conclusions how this phase influences RNA levels, how these levels influence ZIKV infection cannot be made. The general observation that RNA sensors are expressed at different levels in different tissues and that expression in lymphatic tissue is higher than in epithelial tissue does not explain how ZIKV is or is not sensed and what enables it to spread. All in all, the data is not enough to substantiate the conclusions made. Vice versa the data are so extensive that clear correlations cannot be seen. Many of the results could be interpreted differently, mostly because the final experimental prove of "sensing" or "dissemination" or "RNA expression and IFN signaling" is missing and much is left to hypothesizing.

Authors: We respectfully disagree with the reviewer that "How this picture would look in a progesterone-low phase has unfortunately not been studied", as several publications have already addressed this and have shown that vaginal ZIKV infection does not occur in the progesterone-low phase of the estrous cycle independent of innate immunity (Caine et al., 2019; Tang et al., 2016). Our general conclusions that low expression of RNA sensors in the LFRT, in the progesterone-rich phase of the estrous cycle, contributes to the susceptibility of vaginal mucosa to ZIKV infection is supported by expression studies in both mice and macaques, sorting of various LFRT compartments (not just the epithelium), and most importantly we provide genetic proof for these observations using various sensor KO mice. We agree that the interpretation of our data is at times complicated, and we have appreciated all the reviewer comments to help us better articulate these interpretations and more clearly state our conclusions.

However, we disagree with the general assessment that we have not provided enough data to support the conclusions that we have drawn from this study. We hope the additional data provided and the explanations to each of the points below will further clarify how the data presented in literature and here support our general conclusions.

Specific comments:

Abstract: To make the conclusions basal expression basal levels and sensing in DMPA treated vs untreated mice should be checked. And correlated to replication and dissemination.

Authors: We now provide the steady state expression of the sensors under natural estrus versus diestrus compared to DMPA treated animals, and show no significant change in the expression of the various RNA sensors under these conditions, with the only exception being slightly higher expression of *Tlr3* in the LFRT of DMPA-treated animals (Supplemental Fig. 3). In regard to comparison of vaginal ZIKV infection under different estrous conditions, please refer to Tang et al. 2016, where the Shresta group compared vaginal ZIKV infection in diestrus-like versus estrus-like stages and showed that mice in diestrus-like stage succumb to vaginal ZIKV infection, whereas those in estrus-like stage are resistant (Tang et al., 2016). Thus, it is not feasible to measure viral replication and dissemination in estrus-like mice, as they do not become infected after vaginal inoculation. Mechanisms other than level of RNA sensor expression are responsible for resistance to viral infection during the estrus phase, as this stage renders the vaginal mucosa resistant to DNA viral pathogens (HSV) as well.

Line 55 Are macaques more susceptible to ZIKV infection during the progesterone-high phase?

Authors: Yes, DMPA treatment renders the macaques more susceptible to ZIKV vaginal infection as well (Carroll et al., 2017). Similarly, earlier studies have shown susceptibility to vaginal infection with SIV after progesterone treatment (Marx et al., 1996; Smith et al., 2000) or during/following high progesterone phase of the estrous cycle (Kersh et al., 2014).

Line 79 How is this in macaques?

Authors: Please refer to Figure 6 showing ISGs and ZIKV load in different macaque tissues at different time-points.

Fig 2 b is the increase statistically significant?

Authors: We have now included a few more baseline samples for this comparison and can now detect a statistically significant difference in the expression of *Irf7* between baseline and day 4 post infection samples.

Line 123 Which cells are responsible for the *Irf7* increase if the CD11b subset decreases? If those cells leave, have they sensed the virus and migrate to the lymphnodes? Like in Fig 5 the cellular compartments could be checked for *Irf7*. And what about the increase in other sensing markers like cytokines/IFN?

Authors: We have now measured *Irf7* expression in the various cell subsets of the LFRT over time. The data suggests that the epithelial cells at day 4, and stromal and non-PMN cells at day 6 may mainly be responsible for *Irf7* induction. We cannot rule out the possibility that activated CD11b cells that leave the FRT also express some *Irf7*, because in non-PMN cells the expression is higher at day 6 than earlier time-points.

Line 126 Not only CCR2 mediates Monocyte migration.

Authors: We agree, and to make the statement more accurate we replaced “required” with “important”.

Line 133 The main conclusion would be: monocyte migration to the tissue (and *ccr2* signaling) is not needed for sensing. Could be other (resident) cells.

Authors: We have replaced the conclusions with the following statement: “These data suggest that CCR2-mediated monocyte recruitment to the LFRT tissue is not required for anti-ZIKV immunity, and that ZIKV induces moderate activation of existing APCs in the LFRT.”

Line 134 How do these results show that the sensing is dampened and delayed? Or that the APC activation is moderate? Compared to what, other tissues, other cells, other viruses, progesterone-low mice? What is normal?

Authors: Although we can detect viral replication in LFRT within day 2 (Fig 1a), there is only sporadic induction of *Irf7* and about 1.5-fold increase in DC activation marker CD86 starting at day 3 or 4, which shows that viral sensing is dampened and delayed for inhibiting early viral replication. We also draw these

conclusions from the extensive comparison of APC and T cell activation that we have done among different routes of viral infection (Khan et al., 2016; Trapecar et al., 2018). As mentioned earlier, mice in progesterone low or estrous cycle do not become infected with ZIKV; therefore, it is not feasible to compare antiviral innate immune response between diestrus- and estrus-cycle mice.

Line 136 Where is it shown that T cells are primed 4-6 days post infection? Fig 1 only shows 4 dpi, and Fig S2 only shows 6 and 12 dpi, and Fig 2 doesn't show T cell priming?

Authors: We show that viral clearance from LFRT is T-cell dependent (Fig. 1b), and that the frequency of activated T cells increases in the LFRT by around days 6-12 post infection (Fig. 1c). Since APC activation in the iLN is also detected at around day 4 post infection (Fig. 2d) we had concluded that T cells must be primed around days 4-6 post infection. However, to make sure we do not create confusion, we have removed the statement regarding T cell priming and focused on the timing of the viral sensing and APC activation in our conclusions.

Line 146 The RNA copy numbers between KO mice and WT mice do differ about a log.

Authors: We agree that some differences are detected, however, as stated in the revised manuscript, these differences were not statistically significant, despite the large number of animals included in each cohort.

Line 154 "ZIKV replication was strongly inhibited by the presence of sensors " ... indicating that the sensing is not dampened?"

Authors: This statement refers to data shown in Figures 3b-d, which include the UFRT, iLN, and spleen tissues. Indeed, in these "other tissues" viral sensing is NOT dampened.

Fig 2. To conclude that sensing is delayed or dampened, more direct sensing markers need to be evaluated, and a fair control group which shows "normal" sensing included.

Authors: We have extensively shown and described that vaginal infection with RNA viral pathogens results in dampened and delayed viral sensing compared to several other routes of infection including intraperitoneal, subcutaneous, transcervical (Khan et al., 2016), and rectal (Trapecar et al., 2018). In this study we focused on the reason as to why RNA viral sensing is dampened during the susceptible stage of the estrus cycle (high progesterone) and what the consequence of this vulnerability may be for ZIKV dissemination to other tissues.

Line 160 "RNA sensing and IFNAR signaling provide moderate to minimal protection in the LFRT compared to other tissues" How can this be concluded if

the no experiment has been done where other tissues were inoculated with ZIKV and sensing and dissemination monitored there? Isn't simply the site of inoculation the site where replication is most prominent (and sensing and therefore IFN signaling occurring first to minimize spreading)?

Authors: It is already well established that subcutaneous or intraperitoneal infection does not lead to ZIKV replication in WT mice, but it does occur in mice lacking IFN signaling (Dowall et al., 2016; Lazear et al., 2016; Rossi et al., 2016). However, upon vaginal inoculation, viral replication can occur in WT mice at similar magnitude as in the *Ifnar1*^{-/-} or single and double sensor KO mice, with only a 2-3 log increase in the absence of all sensors in TKO mice, thus suggesting viral sensing is dampened at this particular site of infection. Therefore, the site of inoculation can be the site of highest amount of viral replication only if strong sensing and innate immunity does not inhibit viral replication.

Line 164 Again, maybe simply the site of injection decides for the high replication? The protection in other tissues might only be due to delayed viral kinetics (simple time and anatomy) but a preactivated immune system? Vice versa, if the skin/liver/blood would be infected first, would the virus spread faster to the LFRT? Or not at all because there the sensors are more effective?

Authors: These experiments have already been performed in mice (Dowall et al., 2016; Lazear et al., 2016; Rossi et al., 2016) and in macaques (Hirsch et al., 2017; Osuna et al., 2016). In mice, systemic viral replication is inhibited in WT animals but not in *Ifnar1*^{-/-} or *Irf3*^{-/-} *Irf5*^{-/-} *Irf7*^{-/-} mice, demonstrating that viral sensing and IFNAR signaling are both necessary for protection against systemic ZIKV. However, when macaques are systemically inoculated, viral replication occurs in all tissues, including the LFRT, and this is likely because similar to human cells, ZIKV can inhibit INFAR signaling in macaques.

Line 169 TLR3 is marginally enriched

Authors: We have added "marginally" in this sentence.

Line 170 why are the macaques not DMPA treated? Would be more comparable with Fig 6d

Authors: We used archival control macaque tissues to generate this figure, and as those animals were not specifically euthanized for the purposes of this comparison, they were not DMPA treated. Instead, we now provide data comparing the sensor expression in DMPA-treated versus non-DMPA treated mice at either diestrus or estrus phases of the estrous cycle (Supplementary Fig. 3)

Line 176 The conclusion that epithelium has a dampened immunity due to the

fact that it has less sensors expressed than in an (always slightly activated) lymphoid tissue does not seem fair. Which non-lymphatic tissue shows the same activation as lymph node and therefore has “not-dampened” immunity?

Authors: We disagree with the assessment that lymphoid tissue is “always slightly more activated” than the LFRT. For example, we refer the reviewer to Supplemental Figure 1b, where we show a comparison of the fraction of activated T cells in various uninfected (UI) tissues. Clearly, the UI LFRT has a higher fraction of activated T cells than the UI iLN and spleen. Furthermore, we have already shown that infection of the UFRT (Khan et al., 2016) and the colon (Trapeacar et al., 2018) show similar kinetics of immune activation as a lymphoid organ, which highlights the unique features of the LFRT tissue.

Line 186 Only the induction of sensors by different cell compartments is measured. This does not explain which compartment is responsible for the first sensing, simply which compartments responds to a already sensed infection by increasing sensing themselves.

Authors: We agree and certainly did not claim that our analysis meant to measure the kinetics of viral sensing, rather we wanted to determine the contribution of different cellular compartments that sense ZIKV. However, luminal PMNs and the epithelium are likely the first compartments to come in contact with vaginally inoculated ZIKV, followed by stromal and non-PMN immune cells. In fact, this kinetics is somewhat supported by our newly generated *Irf7* induction data attached above in this response letter, showing moderate induction by epithelial cells within day 4, followed by induction in stromal and other immune cells at day 6 post infection.

Line 198 If all compartments show dampened RNA sensor expression, how is this in other tissues?

Authors: This is indeed an interesting question and one that would best be addressed in future studies via single cell RNA sequencing platform.

Line 200 If IFNAR signaling in the LFRT is examined, why not measure IFN levels? Especially IFN III levels and signalling should be focused on as this type is important in mucosal tissues. IFNAR $-/-$ mice of course are a nice tool, but it only focuses on type I and II IFN and might not explain sensing and replication in the FRT.

Authors: We have already shown that type I/III IFNs are minimally induced up to days 3 post vaginal ZIKV infection (Khan et al., 2016). The Diamond group recently showed IFN λ is induced at day 4 post vaginal ZIKV infection and results in some protection during the progesterone-rich stage (Caine et al., 2019). The timing of IFN λ induction is also consistent with the kinetics of when we observe APC activation and *Irf7* induction in the LFRT.

Fig 5f. This Fig like Fig 3a shows that sensing in the FRT in wt mice is strong and reduces replication of the virus.

Authors: Our message is that despite the presence of all the sensors, WT mice still get infected only when the virus is inoculated via vaginal route, but not through other routes. However, if the WT animals are first infected i.p. with LCMV, which results in systemic IFN induction and also induction of the sensors in the LFRT, or if the vaginal epithelium is treated with acitretin to induce RIG-I expression, then the mice are protected from vaginal ZIKV infection (Khan et al., 2016). These observations led us to further investigate why WT mice are susceptible to ZIKV infection only when inoculated via vaginal route. In this study, we measured the expression of the sensors in LFRT compared to other tissues, and found them to be lower in both mice and macaques. We also used a genetic approach to show that while some sensing does occur in the LFRT of WT animals as is evident by the significant differences observed in viral loads between WT and TKO mice in Figures 3a and 5f, sensing is not sufficient to inhibit viral replication in WT mice. Using all this data, we have concluded that the magnitude of viral sensing is much lower in the LFRT compared to the other tissues, as this level of sensing cannot inhibit establishment of infection upon vaginal inoculation.

Line 206 how is the sensing insufficient if the viral loads are decreased?

Authors: While viral replication is somewhat controlled via innate mechanisms early during the infection, as is evidenced by the difference between WT and TKO mice (Fig. 3a and 5f), virus persists in the absence of adaptive immune responses (Fig. 1a-b). These observations demonstrate that despite modest level of sensor expression in the LFRT, viral sensing is not sufficient to control viral replication in the LFRT of WT mice.

Fig 6 Comparing the sensors within the different tissues would be interesting. Which sensors are highest expressed in the different tissues? And then compare uninfected vs infected. And compare the induction of sensors compared to the specific tissue. It looks like the induction in the vaginal tissue is higher than in the ILNs

Authors: In Figure 6 we are already comparing the sensors between the three different tissues by setting the levels in the iLN of uninfected WT mice to 1 and showing the expression or induction of each gene as fold over UI WT iLN. To clarify, in the WT mice, induction of *Irf7* and sensors is higher in the iLN than the LFRT, despite much higher viral loads in the LFRT than the iLN, which supports our conclusions regarding poor sensing of the virus in the LFRT compared to the iLN.

Fig 6d The induction (compared to basal UI levels) of sensors in the LFRT is always higher than in the iLNs

Authors: We agree that this is indeed a complicated data set and we hope the following explanation will help to clarify our conclusions. A certain threshold of viral replication is required to induce the expression of sensors and other ISGs. RNA sensors can be induced either directly in response to signaling downstream of viral sensors or in response to IFNAR signaling, and are thus considered to be ISGs. We show that in the LFRT, viral sensing is more protective than IFNAR signaling, as LFRT viral loads are higher in TKO compared to *Ifnar1*^{-/-} mice (Fig. 3a). Despite high viral loads in all tissues, the sensors/ISGs are not induced in the TKO mice, demonstrating that viral sensing is required for further induction of the sensors. However, the sensors are induced only in the FRT of *Ifnar1*^{-/-} mice but not in their iLN, suggesting that sensor induction can occur downstream of viral sensing in the FRT, but their induction in the iLN requires both viral sensing and IFNAR signaling. Thus, in the WT animals, the transient induction of sensors/ISGs in the iLN is due to transient and low viral loads. In the TKO mice, low induction of sensors/ISGs is due to absence of viral sensing. In *Ifnar1*^{-/-} mice, high viral loads results in high induction of sensors/ISGs in the FRT, but not in iLN, because the induction of these genes in the iLN is dependent on both viral sensing and IFNAR signaling. Interestingly, the induction of the sensors and *Irf7* in macaques mirrors that of *Ifnar1*^{-/-} mice, supporting the conclusion that ZIKV antagonizes IFNAR signaling in macaques, which is why virus disseminates systemically after vaginal inoculation of macaques, but not WT mice.

Line 226-240 this parts needs to be easier to grasp.

Authors: We have further simplified the text, please refer to the revised text in Lines 238-241 in the revised manuscript.

Fig 6E This Figure shows that sensing at the FRT leads to lower dissemination. If sensing or IFN signaling is decreased more virus spreads throughout the system. Doesn't this argue for the fact that the sensing in the FRT is efficient?

Authors: This argues that sensing in the FRT would be sufficient, only if ZIKV cannot antagonize IFNAR signaling once the virus disseminates to other tissues, such as in WT mice. However, we know that ZIKV can interfere with IFNAR signaling in humans, and the very similar pattern of viral dissemination and sensor/ISG induction in macaques and the *Ifnar1*^{-/-} mice suggests that ZIKV can also antagonize IFNAR signaling in macaques.

Line 242 The data does not back up this general conclusion. Collectively, these data demonstrate that ZIKV is already sensed in the FRT, but has the ability to disseminate (in macacaques mybe due to IFN antagonization and in mice only efficiently if IFN or sensing is inactivated).

Authors: We agree, and we have updated the conclusion statement accordingly.

Line 245 “mechanism” is overstated

Authors: We have changed the statement to “our data supports a potential phenomenon”

Line 248 the association seem weak. More other tissues that are not lymphatic are needed.

Authors: We have already included the UFRT as the most relevant non-lymphatic tissue in this study and Figures 3, 4, and 6 all include an analysis of both the LFRT and the UFRT tissues. We also attempted to infect ZIKV rectally; however, due to the canonical innate response that is generated in the rectum, ZIKV was not able to replicate efficiently in WT animals, but it could in *Ifnar1*^{-/-} mice.

Line 250 why the low RNA expression and not the low availability of APCs/ T cells/ B cells? From the data it is not conclusive that the RNA expression is responsible for replication.

Authors: Our conclusion is that low expression of RNA sensors contributes to early viral replication, whereas viral clearance from LFRT eventually relies on T/B cells (Fig 1b). We agree that low availability of immune cells in the LFRT could be a contributor to the low expression of the RNA sensors in the vaginal mucosa, and have clarified this in the statement.

Line 265 could IFN-L induction be measured?

Authors: We have already reported no significant induction of IFN-L up to day 3 post vaginal ZIKV infection (Khan et al., 2016) and the Diamond group recently showed it's induction at day 4 post vaginal ZIKV infection (Caine et al., 2019), which is consistent with when we also detect APC activation and *Irf7* induction.

Line 275 or the fact that ZIKV has been sensed in the LFRT and needs to disseminate from there, which is dampened due to its activation and IFNAR signaling?

Authors: The statement in line 275 is based on the differences that we observe in viral loads between TKO and *Ifnar1*^{-/-} mice in various tissues. Exactly how ZIKV is disseminated from the LFRT to other tissues remains unknown, however, as absence of IFNAR signaling in myeloid cells is sufficient to promote viral dissemination (Tang et al., 2016), it is possible that infected myeloid cells are responsible for viral dissemination.

Line 282 or the disseminating APCs already prime the distal tissue so that the

virus cannot disseminate properly

Authors: Data provided by Tang et al suggests that absence of IFNAR signaling in myeloid cells (*LysM Cre+ Ifnar1^{flx/flx}*) is sufficient to result in systemic viremia upon vaginal inoculation of ZIKV, which also supports the model that we present in this section. It is also possible that in WT mice the migrating immune cells may prime the distal tissues and contribute to inhibition of viral dissemination.

Reviewer #4 (Remarks to the Author):

I appreciate the author's attention to the comments of all four referees that provided comments on the manuscript. Significant changes were made to the manuscript that in many instances improve clarity and context. My appreciation of the work was enhanced by the author's rebuttal and a second look at the paper. No critical issues remain.

Authors: We appreciate this reviewer's support for publication of our revised manuscript.

Caine, E.A., Scheaffer, S.M., Arora, N., Zaitsev, K., Artyomov, M.N., Coyne, C.B., Moley, K.H., and Diamond, M.S. (2019). Interferon lambda protects the female reproductive tract against Zika virus infection. *Nat Commun* 10, 280.

Carroll, T., Lo, M., Lanteri, M., Dutra, J., Zarbock, K., Silveira, P., Rourke, T., Ma, Z.M., Fritts, L., O'Connor, S., *et al.* (2017). Zika virus preferentially replicates in the female reproductive tract after vaginal inoculation of rhesus macaques. *PLoS Pathog* 13, e1006537.

Dowall, S.D., Graham, V.A., Rayner, E., Atkinson, B., Hall, G., Watson, R.J., Bosworth, A., Bonney, L.C., Kitchen, S., and Hewson, R. (2016). A Susceptible Mouse Model for Zika Virus Infection. *PLoS neglected tropical diseases* 10, e0004658.

Hirsch, A.J., Smith, J.L., Haese, N.N., Broeckel, R.M., Parkins, C.J., Kreklywich, C., DeFilippis, V.R., Denton, M., Smith, P.P., Messer, W.B., *et al.* (2017). Zika Virus infection of rhesus macaques leads to viral persistence in multiple tissues. *PLoS Pathog* 13, e1006219.

Kersh, E.N., Henning, T., Vishwanathan, S.A., Morris, M., Butler, K., Adams, D.R., Guenthner, P., Srinivasan, P., Smith, J., Radzio, J., *et al.* (2014). SHIV susceptibility changes during the menstrual cycle of pigtail macaques. *Journal of medical primatology* 43, 310-316.

Khan, S., Woodruff, E.M., Trapecar, M., Fontaine, K.A., Ezaki, A., Borbet, T.C., Ott, M., and Sanjabi, S. (2016). Dampened antiviral immunity to intravaginal

exposure to RNA viral pathogens allows enhanced viral replication. *J Exp Med* 213, 2913-2929.

Lazear, H.M., Govero, J., Smith, A.M., Platt, D.J., Fernandez, E., Miner, J.J., and Diamond, M.S. (2016). A Mouse Model of Zika Virus Pathogenesis. *Cell Host Microbe* 19, 720-730.

Marx, P.A., Spira, A.I., Gettie, A., Dailey, P.J., Veazey, R.S., Lackner, A.A., Mahoney, C.J., Miller, C.J., Claypool, L.E., Ho, D.D., *et al.* (1996). Progesterone implants enhance SIV vaginal transmission and early virus load. *Nat Med* 2, 1084-1089.

Osuna, C.E., Lim, S.Y., Deleage, C., Griffin, B.D., Stein, D., Schroeder, L.T., Orange, R., Best, K., Luo, M., Hraber, P.T., *et al.* (2016). Zika viral dynamics and shedding in rhesus and cynomolgus macaques. *Nat Med* 22, 1448-1455.

Rossi, S.L., Tesh, R.B., Azar, S.R., Muruato, A.E., Hanley, K.A., Auguste, A.J., Langsjoen, R.M., Paessler, S., Vasilakis, N., and Weaver, S.C. (2016). Characterization of a Novel Murine Model to Study Zika Virus. *Am J Trop Med Hyg* 94, 1362-1369.

Smith, S.M., Baskin, G.B., and Marx, P.A. (2000). Estrogen protects against vaginal transmission of simian immunodeficiency virus. *J Infect Dis* 182, 708-715.

Tang, W.W., Young, M.P., Mamidi, A., Regla-Nava, J.A., Kim, K., and Shresta, S. (2016). A Mouse Model of Zika Virus Sexual Transmission and Vaginal Viral Replication. *Cell Rep* 17, 3091-3098.

Trapeacar, M., Khan, S., Cohn, B.L., Wu, F., and Sanjabi, S. (2018). B cells are the predominant mediators of early systemic viral dissemination during rectal LCMV infection. *Mucosal Immunol* 11, 1158-1167.

Reviewers' comments:

Reviewer #3 (Remarks to the Author):

The results in the study by S Khan et al are interesting but leave much room for interpretation which is mainly due to the fact that the analyzed system is highly complex. The main findings and conclusions should be described more clearly and it should also be stated what they don't show and what is speculation. Especially, this study does not experimentally address the difference between the progesterone high and low setting, but only analyses effects during a progesterone high setting, and therefore does not give an explanation to why the FRT is more ZIKV susceptible during the diestrous phase, which should be made very clear to the reader.

Line 60 "However, we have very limited knowledge about why innate immunity is uniquely dampened against ZIKV and other RNA viral pathogens in the LFRT during progesterone-high conditions "

This is also a question that is not answered in your study.

Fig S3. Why does TLR3 expression under DMPA treatment not reflect the Diestrous phase?

As Fig S3 shows that the expression of innate sensors under estrous and diestrous phases is very similar, how can you conclude that the RNA sensing are reduced upon progesterone high phases? Where do you show that this RNA sensing is higher during progesterone low phases?

Line 179 Together, these data suggest that most RNA sensors are not hormonally regulated and their low basal expression in the FRT likely contribute to the observed dampened innate immunity to ZIKV in mice.

If not hormonally regulated, why is there a difference between virus infection in mice during diestrous, or estrous phase? So why is DMPA treatment needed, if the hormones don't change anything? If the low basal expression contributes to dampened immunity, why don't you see FRT-replication in non DMPA treated mice, where there is also low expression?

Fig 2 how would "non-delayed" sensing look like? What is the control?

Line 254 “Our data supports a model in which, during the high progesterone phase of the estrous cycle or during early pregnancy, the low expression of RNA sensing PRRs, which can be due to low abundance of immune cells in the LFRT, contributes to high viral replication in the vaginal mucosa.”

Please specify. It sounds like high progesterone correlates with low RNA expression, which it doesn't (Fig S3). Please explain what changes under high progesterone levels compared to low progesterone levels, according to your model.

Lin 357. Can you speculate why under high progesterone conditions it is more susceptible? As your data indicates it is not due to RNA sensing?

It might confuse the reader that there often is an emphasis on “the high progesterone phase”, which suggests that the difference between high and low will be studied. However, simply the whole setting is “progesterone high” so conclusions about “why innate immunity is uniquely dampened ... during progesterone-high conditions” can of course not be made. This should become clearer. Maybe state that you only study during the high susceptibility phase, and discuss how this might differ from the progesterone low state.

Rebuttal

“As mentioned earlier, mice in progesterone low or estrous cycle do not become infected with ZIKV; therefore, it is not feasible to compare antiviral innate immune response between diestrus- and estrus-cycle mice.”

Would this then be the perfect control, because here sensing should be high?

“In this study we focused on the reason as to why RNA viral sensing is dampened during the susceptible stage of the estrus cycle (high progesterone) and what the consequence of this vulnerability may be for ZIKV dissemination to other tissues.”

As you did not include a low progesterone control you did not examine a difference of sensing between estrus cycle stages. After reading the manuscript I do not know why it is dampened, simply what might contribute.

The results in the study by S Khan et al are interesting but leave much room for interpretation which is mainly due to the fact that the analyzed system is highly complex. The main findings and conclusions should be described more clearly and it should also be stated what they don't show and what is speculation. Especially, this study does not experimentally address the difference between the progesterone high and low setting, but only analyses effects during a progesterone high setting, and therefore does not give an explanation to why the FRT is more ZIKV susceptible during the diestrus phase, which should be made very clear to the reader.

Authors: We thank the reviewer for their persistence to make sure that our message is clear, which also remains our ultimate goal. We agree that the female reproductive tract immunology is complex mainly due to the biological changes that occur due to hormonal fluctuations during the various stages of the estrous cycle ¹. Understanding how exactly changes in estradiol and progesterone affect susceptibility to various sexually transmitted pathogens is an active area of ongoing investigation; however, this was not the focus of our current study and we apologize if we have not been clear about this. There are numerous publications and experimental evidence demonstrating that mice, non-human primates, and humans are less susceptible to vaginal transmission of pathogens during the estradiol high phase of the estrous cycle. This difference in susceptibility between estradiol-high and progesterone-high phases is due to factors that are not necessarily related to pathogen sensing, as it occurs similarly for bacteria (i.e. Chlamydia), DNA (i.e. HSV), and RNA (i.e. HIV, ZIKV, LCMV) viral pathogens. Importantly, this difference in susceptibility is likely due to physical mucosal barrier inhibition, such as changes in estradiol-induced vaginal mucous that inhibits antigen penetration ², as well as differences in genital mucosal permeability due to changes in epithelial thickness and expression of cell-cell adhesion molecules ³. DMPA, a progesterone-based contraceptive, is commonly used to make animals susceptible to vaginal infection by inducing similar changes to the mucous and vaginal epithelium that also naturally occur during diestrus. However, during early pregnancy, vaginal ZIKV infection can still occur in the absence of DMPA treatment, and this coincides with high progesterone levels at early pregnancy. Therefore, we opted to categorize all these conditions, including DMPA-treated animals, diestrus phase, and early pregnancy into "progesterone-high" conditions, and this has unfortunately created some confusion. We completely agree that we have not compared progesterone-high to progesterone-low in this study, as we do not believe a progesterone-low condition is relevant, since the animals do not become infected due to mucosal barrier reasons discussed above. To rectify this, we have explained in the introduction that we are using a progesterone high condition, and have removed most of the reference to "progesterone-high" conditions throughout the rest of the main text.

Furthermore, in this study, our goal was to build on our previous findings that in contrast to HSV vaginal infection (also in DMPA-treated mice), which elicit a strong type I IFN response ^{4, 5, 6}, when we infected DMPA-treated mice with either LCMV or ZIKV, we detected highly dampened type I IFN expression ⁷. In contrast, if we infect mice trans-cervically ⁷, or intra-rectally ⁸, we indeed observed a strong type I IFN response, which also led to rapid APC activation. We had also shown that if RNA sensor expression is induced in the vaginal mucosa (either by applying acitretin or by systemically infecting the animals prior to vaginal infection, where systemic type I IFNs act on all tissues to induce sensor expression) we could inhibit vaginal ZIKV infection ⁷. All of these observations led us to hypothesize that low basal expression of RNA viral sensors in the vaginal mucosa may be contributing to dampened innate immune response against RNA viral pathogens, including ZIKV. For this study, we then used RNA sensor KO and *Ifnar1*^{-/-} animals (all in DMPA-treated mice) to show while RNA sensors provide minimal protection in the vaginal mucosa, this level of protection is not sufficient to inhibit viral replication. In contrast, when virus disseminates to other tissues, viral replication is quickly inhibited when the sensors are present. Therefore, our conclusion is that if the virus gets through the mucosal barrier due to the high progesterone condition of the host, the low expression of RNA sensors contributes to the dampened innate immunity that is observed in the vaginal mucosa. Furthermore, in the absence of IFNAR signaling (*Ifnar1*^{-/-} mice) or if ZIKV can inhibit IFNAR signaling (as it likely does in humans and in NHPs), the vaginal infection can lead to systemic viral dissemination. Thus, we have reasoned from these observations that if vaginal ZIKV infection occurs during early pregnancy in humans, high levels of viral replication in the vaginal tissue can lead to systemic infection, which can have adverse consequences for the fetus.

We have made an effort to clarify all these points throughout the manuscript.

Line 60 “However, we have very limited knowledge about why innate immunity is uniquely dampened against ZIKV and other RNA viral pathogens in the LFRT during progesterone-high conditions “

This is also a question that is not answered in your study.

Authors: As discussed above, our intention in this study was not to compare progesterone-low (infection simply does not occur due to intact mucosal barriers) to progesterone-high conditions, but to emphasize that we only studied the progesterone-high condition by DMPA treating the mice. We apologize for the confusion our wording has caused and we have now addressed this in the main text.

Fig S3. Why does TLR3 expression under DMPA treatment not reflect the

Diestrus phase?

Authors: Indeed, we saw this difference reproducibly in two independent experiments, and our data is also consistent with results from the Rosenthal group, where they also observed a significant increase in TLR3 and TLR5 expression in whole vaginal tissue when animals were Depo-treated⁹. While the exact reason as to why TLR3 expression under DMPA treatment does not reflect the expression observed under diestrus phase is unknown, it is well appreciated that DMPA also binds to other steroid receptor family of proteins other than progesterone receptor (PR)¹⁰, such as glucocorticoid receptor (GR)¹¹, and can thus have biological consequences that differ from progesterone.

As Fig S3 shows that the expression of innate sensors under estrous and diestrus phases is very similar, how can you conclude that the RNA sensing are reduced upon progesterone high phases? Where do you show that this RNA sensing is higher during progesterone low phases?

Authors: Our intention was not to compare progesterone high to progesterone low phase, but rather understand why vaginal mucosa uniquely elicits a dampened innate immune response to ZIKV under conditions when vaginal mucosa is susceptible to pathogenic infections, which only occurs during progesterone high conditions. We have now clarified this point in the introduction as the rationale for treating the mice with DMPA prior to vaginal infections.

Line 179 Together, these data suggest that most RNA sensors are not hormonally regulated and their low basal expression in the FRT likely contribute to the observed dampened innate immunity to ZIKV in mice.

If not hormonally regulated, why is there a difference between virus infection in mice during diestrus, or estrus phase? So why is DMPA treatment needed, if the hormones don't change anything? If the low basal expression contributes to dampened immunity, why don't you see FRT-replication in non DMPA treated mice, where there is also low expression?

Authors: As discussed above, progesterone and DMPA cause many changes to the vaginal mucosa that are only partially related to sensor expression¹. Briefly, DMPA treatment is needed to alter the physical mucosal barrier, such as mucus, epithelial thickness, and expression of cell-cell adhesion molecules to allow pathogen penetration and ultimately infection of the vaginal tissue. However, although hormonal changes do not change the expression of most of the RNA sensors, the low basal expression of RNA sensors results in dampened innate immunity, which ultimately results in enhanced viral replication in vaginal mucosa compared to other tissues. This innate viral escape mechanism is only evident under conditions where viral mucosal penetration can occur, such as in diestrus phase, DMPA-treated, or early pregnancy, which are all progesterone-high

conditions. Our conclusion is further supported by Caine et al. where using ovariectomized mice they showed that IFN-L has moderate antiviral effect against vaginal ZIKV infection only if mice are treated with progesterone, but not in mice treated with estrogen or progesterone plus estrogen (please see attached figures below that also show examples of physical changes caused by the hormones) ¹².

We have modified the relevant text in the manuscript and have added further explanation in the discussion section to clarify our conclusion.

Figure 2A (left) and S3 (right) from Caine EA et al., Nat Commun 2019

Fig 2 how would “non-delayed” sensing look like? What is the control?

Authors: We know the sensing is delayed because we have extensively evaluated the kinetics of APC activation when the same RNA viral pathogen (LCMV) is inoculated via different routes ^{7,8}. We measured APC activation in DMPA-treated mice that were either vaginally (i.vag.) or trans-cervically (t.c.) infected with LCMV. While t.c. infection resulted in rapid activation (days 1 and 2 post infection) of Ly6c^{hi} IAIE⁻ and Ly6c⁺ IAIE⁺ APCs as measured by CD86 and CD40 upregulation, i.vag. infection did not result in significant change in APC activation up to day 3 p.i. (Top panel shown here and Fig 6G from Khan et al. JEM 2016). Similar to t.c. infection, when animals were intra-rectally infected with LCMV, rapid APC recruitment and activation was observed in the rectal mucosa (Lower panel shown here and Fig 4b from Trapecar et al. Mucosal Immunol 2018). We now show that after i.vag. ZIKV infection, activation of APCs is observed around day 4 p.i., thus our claim that sensing is “delayed” in the vaginal mucosa. We and Akiko Iwasaki’s group (Yockey et al, Cell 2016) have attempted

to infect WT mice with ZIKV via other routes of infection, but failed to detect viral replication, likely due to the more robust innate immunity that is elicited when ZIKV is inoculated via these other routes. This is indeed in support of our conclusion that innate immunity is uniquely dampened in the vaginal mucosa.

After vaginal versus intra-cervical infection with LCMV (Khan et al. JEM 2016, Figure 6G).

After Rectal infection with LCMV (Trapezar et al. Mucosal Immunol 2018, Figure 4b).

In the context of this study alone, we have provided additional explanation explaining that the kinetics of viral sensing and innate immune activation is slower than viral replication, and as a result innate sensing is not sufficient to completely inhibit early viral replication.

Line 254 “Our data supports a model in which, during the high progesterone phase of the estrous cycle or during early pregnancy, the low expression of RNA sensing PRRs, which can be due to low abundance of immune cells in the LFRT, contributes to high viral replication in the vaginal mucosa.”

Please specify. It sounds like high progesterone correlates with low RNA expression, which it doesn't (Fig S3). Please explain what changes under high progesterone levels compared to low progesterone levels, according to your model.

Authors: Thank you for pointing this out, our intention was to emphasize that this becomes relevant under high progesterone conditions, and we can certainly see how this sentence could be interpreted in different ways. We have altered the text and added more explanation to convey our message more clearly.

Lin 357. Can you speculate why under high progesterone conditions it is more

susceptible? As your data indicates it is not due to RNA sensing?

Authors: As discussed above, we have now provided an explanation for this in the Introduction and in the Discussion sections.

It might confuse the reader that there often is an emphasis on “the high progesterone phase”, which suggests that the difference between high and low will be studied. However, simply the whole setting is “progesterone high” so conclusions about “why innate immunity is uniquely dampened ... during progesterone-high conditions“ can of course not be made. This should become clearer. Maybe state that you only study during the high susceptibility phase, and discuss how this might differ from the progesterone low state.

Authors: We agree that our current phrasing has created much confusion, and we have now emphasized that we are only studying the high-progesterone phase, as that is the relevant phase when viral infection can actually occur due to changes to the mucosal barrier (as discussed above).

Rebuttal

“As mentioned earlier, mice in progesterone low or estrous cycle do not become infected with ZIKV; therefore, it is not feasible to compare antiviral innate immune response between diestrus- and estrus-cycle mice.”

Would this then be the perfect control, because here sensing should be high?

Authors: Other physical barriers such as the mucous and epithelial integrity contribute to protection against all pathogens during the estradiol-high phase, and we are certainly not claiming that sensing is the protective mechanism during this phase. As animals cannot be infected during the estrus phase of the cycle, for reasons other than level of sensor expression, we do not believe this is a relevant control for these studies.

We have already shown that if we artificially increase the expression of the sensors in the LFRT either by applying acitretin locally or by systemically infecting the animals with LCMV to increase sensor expression in all tissues, we can inhibit vaginal ZIKV infection ⁷. More convincingly, we now show only about 2 log difference in viral replication in the LFRT between WT and TKO mice, which provides genetic evidence that the natural level of sensors present in the LFRT are not sufficient to inhibit viral replication in the LFRT, as they do in all other tissues (UFRT, iLN, spleen) where no viral replication is detected in WT mice.

“In this study we focused on the reason as to why RNA viral sensing is dampened during the susceptible stage of the estrus cycle (high progesterone) and what the consequence of this vulnerability may be for ZIKV dissemination to other tissues.”

As you did not include a low progesterone control you did not examine a difference of sensing between estrus cycle stages. After reading the manuscript I do not know why it is dampened, simply what might contribute.

Authors: Here, we meant to emphasize that we only studied viral infection during the susceptible stage of the estrous cycle, which coincides with high progesterone levels. We have now clarified the language throughout the manuscript to better emphasize this point.

In this study we show that low expression of RNA sensors contributes to the ability of ZIKV to highly replicate in the vaginal mucosa, and that in the absence of IFNAR signaling, virus can disseminate systemically. However, we do not know why RNA sensor expression is low in the vaginal mucosa. While this is not hormonally regulated, it may still be related to reproductive biology and the interesting observations that semen and semen exosomes contain high amounts of RNA^{13,14}. Thus, the low level of RNA sensors in the LFRT may be an evolutionary mechanism to allow reproduction without eliciting excessive inflammatory response to semen RNA, which leaves a window of opportunity for RNA viral pathogens to be sexually transmitted. We have now included a brief discussion about all this in the Discussion section. We hope that our collective findings will spark more interest in the community to begin to address the exact mechanism by which RNA sensor expression is low in the LFRT.

1. Wira, C.R., Rodriguez-Garcia, M. & Patel, M.V. The role of sex hormones in immune protection of the female reproductive tract. *Nat Rev Immunol* **15**, 217-230 (2015).
2. Seavey, M.M. & Mosmann, T.R. Estradiol-induced vaginal mucus inhibits antigen penetration and CD8(+) T cell priming in response to intravaginal immunization. *Vaccine* **27**, 2342-2349 (2009).
3. Quispe Calla, N.E. *et al.* Medroxyprogesterone acetate and levonorgestrel increase genital mucosal permeability and enhance susceptibility to genital herpes simplex virus type 2 infection. *Mucosal Immunol* (2016).
4. Oh, J.E. *et al.* Dysbiosis-induced IL-33 contributes to impaired antiviral immunity in the genital mucosa. *Proc Natl Acad Sci U S A* **113**, E762-771 (2016).
5. Lund, J.M., Linehan, M.M., Iijima, N. & Iwasaki, A. Cutting Edge: Plasmacytoid dendritic cells provide innate immune protection against mucosal viral infection in situ. *Journal of immunology* **177**, 7510-7514 (2006).

6. Shen, H. & Iwasaki, A. A crucial role for plasmacytoid dendritic cells in antiviral protection by CpG ODN-based vaginal microbicide. *J Clin Invest* **116**, 2237-2243 (2006).
7. Khan, S. *et al.* Dampened antiviral immunity to intravaginal exposure to RNA viral pathogens allows enhanced viral replication. *J Exp Med* **213**, 2913-2929 (2016).
8. Trapecar, M., Khan, S., Cohn, B.L., Wu, F. & Sanjabi, S. B cells are the predominant mediators of early systemic viral dissemination during rectal LCMV infection. *Mucosal Immunol* **11**, 1158-1167 (2018).
9. Yao, X.D., Fernandez, S., Kelly, M.M., Kaushic, C. & Rosenthal, K.L. Expression of Toll-like receptors in murine vaginal epithelium is affected by the estrous cycle and stromal cells. *Journal of Reproductive Immunology* **75**, 106-119 (2007).
10. Hapgood, J.P., Kaushic, C. & Hel, Z. Hormonal Contraception and HIV-1 Acquisition: Biological Mechanisms. *Endocr Rev* **39**, 36-78 (2018).
11. Govender, Y. *et al.* The injectable-only contraceptive medroxyprogesterone acetate, unlike norethisterone acetate and progesterone, regulates inflammatory genes in endocervical cells via the glucocorticoid receptor. *PLoS One* **9**, e96497 (2014).
12. Caine, E.A. *et al.* Interferon lambda protects the female reproductive tract against Zika virus infection. *Nat Commun* **10**, 280 (2019).
13. Vojtech, L. *et al.* Exosomes in human semen carry a distinctive repertoire of small non-coding RNAs with potential regulatory functions. *Nucleic Acids Res* **42**, 7290-7304 (2014).
14. Chen, Q., Yan, W. & Duan, E. Epigenetic inheritance of acquired traits through sperm RNAs and sperm RNA modifications. *Nat Rev Genet* **17**, 733-743 (2016).